# Stirring across the Antarctic Circumpolar Current's Southern Boundary at the Greenwich Meridian, Weddell Sea

Ria Oelerich[1], Karen J. Heywood[1], Gillian M. Damerell[1,4], Marcel du Plessis[2], Louise C. Biddle[2], and Sebastiaan Swart[2, 3]

[1]Centre for Ocean and Atmospheric Sciences, School of Environmental Sciences, University of East Anglia, Norwich, NR4 7TJ, United Kingdom
[2]Department of Marine Sciences, University of Gothenburg, Gothenburg, Sweden
[3]Department of Oceanography, University of Cape Town, Rondebosch, South Africa
[4]now at: Geophysical Institute, University of Bergen, and Bjerknes Centre for Climate Research, Bergen, Norway

**Correspondence:** Ria Oelerich (riaoelerich@gmail.com)

**Abstract.** At the Southern Boundary of the Antarctic Circumpolar Current (ACC), relatively warm ACC waters encounter the colder waters surrounding Antarctica. Strong density gradients across the Southern Boundary indicate the presence of a frontal jet and are thought to modulate the southward heat transport across the front. In this study, the Southern Boundary in the Weddell Sea sector at the Greenwich Meridian is surveyed for the first time in high resolution over 2 months during an austral
summer with underwater gliders occupying a transect across the front on five occasions. The five transects show that the frontal structure (i.e., hydrography, velocities and lateral density gradients) varies temporally. The results demonstrate significant, transient (a few weeks) variability of the Southern Boundary and its frontal jet in location, strength and width. A mesoscale cold-core eddy is identified to disrupt the Southern Boundary's frontal structure and strengthen lateral density gradients across the front. The front's barrier properties are assessed using mixing length scales and potential vorticity to establish the cross-
frontal exchange of properties between the ACC and the Weddell Gyre. The results show that stronger lateral density gradients caused by the mesoscale eddy strengthen the barrier-like properties of the front through reduced mixing length scales and pronounced gradients of potential vorticity. In contrast, the barrier-like properties of the Southern Boundary are reduced when no mesoscale eddy is influencing the density gradients across the front. Using satellite altimetry, we further demonstrate that the barrier properties over the past decade have strengthened as a result of increased meridional gradients of absolute dynamic
topography and increased frontal jet speeds in comparison to previous decades. Our results emphasise that locally- and rapidly-changing barrier properties of the Southern Boundary are important to quantify the cross-frontal exchange, which is particularly relevant in regions where the Southern Boundary is located near the Antarctic shelf break (e.g. in the West Antarctic sector).

## 1 Introduction

The Southern Ocean hosts one of the largest current systems on earth, the Antarctic Circumpolar Current (ACC). The eastward
flow of the ACC circulates the Southern Ocean's major source of heat, Circumpolar Deep Water (CDW), and is characterised by strongly tilted isopycnals shoaling poleward (Orsi et al., 1995). Traditionally, the ACC is described with the three major

deep reaching fronts representing boundaries between zones with distinct water mass properties. The seasonal and interannual variability of transport, extent and location of these fronts have been studied extensively over past decades using water mass properties (e.g., Orsi et al., 1995; Kim and Orsi, 2014), fixed sea surface height (SSH) contours, gradients of SSH and mean transport positions (e.g., Sokolov and Rintoul, 2007, 2009a, b; Billany et al., 2010; Gille, 2014; Gille et al., 2016). Enhanced density gradients across the fronts support strong oceanic jets that form the main contribution to the ACC transport and act as barriers to cross-frontal mixing (Naveira-Garabato et al., 2011; Thompson and Sallée, 2012; Chapman and Sallée, 2017). In some studies, the traditional three-front view of the ACC has been expanded by including the Southern Boundary of the ACC to the south (e.g., Billany et al., 2010). However, its definition is not based on the characteristics of a dynamical front (Talley et al., 2011), but rather as a boundary of water mass properties that separates warm ACC waters from colder water masses further south (Orsi et al., 1995). Therefore, the Southern Boundary is often not considered as part of the ACC (Sokolov and Rintoul, 2007) and its changing properties that can enhance or suppress cross-frontal mixing have not been studied extensively in the past.

The focus of this study is the Southern Boundary of the ACC (Fig. 1a), which is climatologically defined as the southern-most limit of Upper Circumpolar Deep Water (UCDW, $\Theta > 1.5°C$ and $S > 34.5$, (Orsi et al., 1995)). The proximity of the Southern Boundary to the continental shelf break varies around Antarctica, where its northernmost displacements are located in areas of cyclonic gyres with clockwise surface circulation in the Weddell and Ross Seas. Specifically in areas where the Southern Boundary is located close to the continental shelf, such as in the West Antarctic Sector, it is considered to play an important role in processes that can aid or oppose the influx of warm waters onto the continental shelf (e.g., Dinniman and Klinck, 2004; Jenkins and Jacobs, 2008; Martinson and McKee, 2012). The frontal jets of the ACC are often seen as barriers to meridional horizontal mixing (e.g., Naveira-Garabato et al., 2011; Thompson and Sallée, 2012; Chapman and Sallée, 2017; Chapman et al., 2020).

The area of interest in this study is located at the Greenwich Meridian in the northern Weddell Sea, where previous studies by Billany et al. (e.g. 2010); Swart et al. (e.g. 2010) have located the Southern Boundary of the ACC at about 55.5°S. Although the Southern Boundary was originally defined as a water mass boundary rather than a dynamical front (Orsi et al., 1995), a more recent study clearly showed that the Southern Boundary is associated with a frontal jet at the Greenwich Meridian (Swart et al., 2010). Thus the Southern Boundary represents the southernmost of the ACC frontal jets at the Greenwich Meridian and marks the boundary between the northern limit of sea ice formation and the ACC.

Previous studies have shown that the mean positions of the major ACC fronts have not shifted southward in response to southward migrating, intensifying westerly winds due to recent climate change (e.g., Chapman et al., 2020; Gille, 2014; Shao et al., 2015; Gille et al., 2016). However, analysed in situ observations, historical reconstructions of ocean conditions, ensembles of coupled climate model simulations and idealised experiments have shown that the ACC's core eastward flow (at 52°S) has accelerated over the past decade (Shi et al., 2021). The acceleration in eastward flow of the ACC was not attributed to changes

in wind strength, but rather to intensifying meridional density gradients in response to upper ocean warming. Nonetheless, satellite altimetry and eddy resolving models suggested an intensifying eddy field within the ACC over the past decade in response to the long-term increase in westerly winds (e.g., Meredith and Hogg, 2006; Hogg et al., 2015; Patara et al., 2016).

60  Studies have shown that mesoscale eddies across the ACC fronts sharpen density gradients and thus strengthen the frontal jets (Williams et al., 2007; Hughes and Ash, 2001), which in turn act to suppress the mixing across the ACC fronts and greatly reduce the meridional exchange of properties, such as heat and carbon (Naveira-Garabato et al., 2011). As a consequence, regions where the ACC fronts have weaker frontal jets, such as downstream of large bathymetric features, are characterised by less suppressed mixing across fronts and thus elevated meridional exchange of properties (e.g., Naveira-Garabato et al.,

65  2011; Thompson and Sallée, 2012). The majority of the aforementioned studies almost entirely focused on the mean positions, transports and barrier properties of the major ACC fronts, whereas processes and dynamics affecting the frontal structure and the frontal jet of the Southern Boundary and meridional exchange of properties across it are poorly understood.

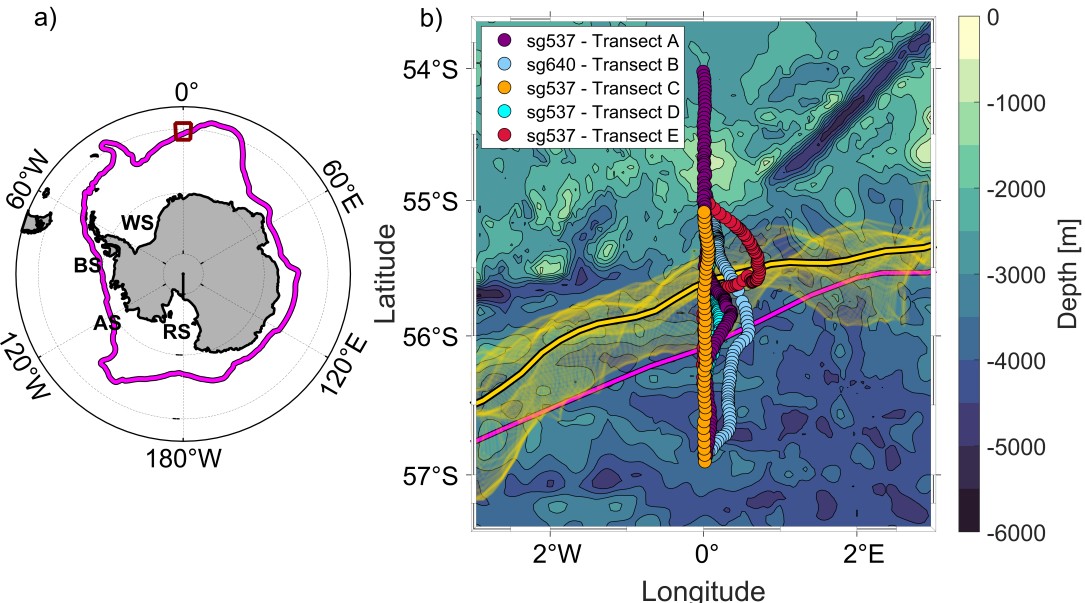

**Figure 1.** (a) Map of the Southern Ocean with the study region outlined by a red box and the climatological mean location of the Southern Boundary (magenta contour in (a) and (b)) derived from water mass properties by Orsi et al. (1995)). (b) Five glider transects superimposed on the bathymetry (Schaffer et al., 2019). Each coloured dot represents the glider's position at the surface following a dive. The -1.16 m contour of absolute dynamic topography from satellite altimetry is shown as a mean over the observational time period (18th October 2019 to 18th February 2020, bold yellow) and daily over the same time period (transparent yellow). The significance of this contour is discussed in the main text in section 2. Key geographic features in (a) are labelled: Ross Sea (RS), Amundsen Sea (AS), Bellingshausen Sea (BS) and Weddell Sea (WS).

In this study, the Southern Boundary's frontal characteristics, barrier/blender properties and short-term variability in the northern Weddell Sea are investigated. We specifically highlight the impacts of mesoscale eddies on the frontal structure of the

Southern Boundary and test the hypothesis that eddies interacting with the Southern Boundary affect density gradients, frontal jet intensity, mixing length scales and mixing across the Southern Boundary. For our analysis we use repeat glider transects crossing the Southern Boundary at the Greenwich Meridian (Fig. 1b) and satellite altimetry to: (i) describe the Southern Boundary's frontal structure and frontal jet intensity, (ii) identify the location, rotational direction and dynamics of mesoscale eddies interacting with the Southern Boundary and (iii) establish how eddies impact the Southern Boundary's barrier/blender properties. We further investigate changes of the Southern Boundary's location and frontal jet intensity using AVISO satellite altimetry from 1993 to 2020 and discuss the potential implications for the barrier/blender properties of the front as well as impacts on sea ice extent. This study is organised as follows: section 2 introduces the Southern Boundary's frontal structure and its variability using the five glider transects. Section 3 describes and quantifies the effects of a mesoscale eddy on the frontal structure using two glider transects (transects A and C, Fig. 1b) . Section 4 evaluates mixing length scales and barrier properties of the Southern Boundary. Section 5 provides the main conclusions and offers suggestions for future work.

## 2   Frontal Structure of the Southern Boundary

As part of the 'Robotic Observations And Modelling in the Marginal Ice Zone' (ROAM-MIZ, www.roammiz.com, Swart et al., 2020) project, two Seagliders (SG537 and SG640) were deployed at the Greenwich Meridian in the northeastern Weddell Sea (SG537: 0.00°W and -55°S, SG640: 0.02°W and -55.01°S) and obtained a total of five repeated crossings of the Southern Boundary (Fig. 1, transects A-E). The average time taken to complete one crossing of the front was 16.6 days (transect A - 6th to 29[th] Nov (321.85 km); transect B - 12[th] Nov to 3[th] Dec (211.90 km), transect C - 29[th] Nov to 7th Dec (202.13 km); transect D - 7[th] Dec to 22[th] Dec (145.56 km) and transect E - 22[nd] Dec to 6[th] Jan (152.42 km). All derived ocean properties use the TEOS10 equation of state (IOC et al., 2010) and therefore temperature will refer to conservative temperature [°C] and salinity will refer to absolute salinity [$g\,kg^{-1}$] throughout this study. We map the profiles of temperature and salinity to a regular 2 m spacing in the vertical by binning the data and then taking the mean value in each bin (Fig. 2). The gliders also provide an estimate of the currents experienced by the gliders during each dive, the dive average current (DAC). Independently from the DAC, we further provide an estimate of the surface currents by calculating the surface drift of the glider during communication with the satellite at the surface. The glider transects are typically oriented cross-front, so that the cross-transect (eastward) velocities capture the majority of the flow associated with the Southern Boundary. We neglect any meridional component of the flow associated with the Southern Boundary at this location. The data are further mapped in the horizontal to a regular 5 km spacing by binning the data in 5 km bins and taking the mean value in each bin. The 5 km interval is chosen to ensure that there are at least two data points per grid point at each depth for the binning process. A uniform grid for all glider transects is necessary to calculate a mean section and standard deviation to highlight areas across the Southern Boundary characterised by strongest variability of water mass properties. The uniform grid further ensures consistent smoothing parameters for the mixing length scale diagnostics introduced in section 4. The absolute along-stream geostrophic velocities are calculated by referencing the geostrophic shear to the eastward component of the DAC. Note that the DAC is linearly interpolated onto the uniform grid for referencing the geostrophic shear. The geostrophic velocities are then horizontally smoothed with a 15 km moving mean

filter, which corresponds to the Rossby radius of deformation (e.g. Chelton et al., 1998) of the region of interest, to reduce the effects of aliasing processes smaller than geostrophic scales.

This study uses the daily satellite-altimetry-derived global sea level data product (SEALEVEL_GLO_PHY_L4_MY_008_047) provided by the Copernicus Marine Environment Monitoring Service (CMEMS) with a horizontal resolution of 0.25x0.25° (approximately 28 km x 16 km in the study region). It covers the period from 1993 onward (DOI: https://doi.org/10.48670/moi-00148, 2022) incorporating all altimetry-carrying Copernicus missions (Sentinel-6A, Sentinel-3A/B) and other collaborative missions (e.g.: Jason-3, Saral[-DP]/AltiKa, Cryosat-2, OSTM/Jason2, Jason-1, Topex/Poseidon, Envisat, GFO, ERS-1/2, Haiyang-2A/B) (Pujol, 2022). Typically, most recent products are available with a 10-month delay. Absolute dynamic topography (ADT) representing sea surface height above the geoid, sea level anomalies (SLA) and surface geostrophic currents up to December 2020 are used in this study. Note that the SLA provided by the altimetry are relative to the 20-year mean from 1993 to 2012.

All glider transects (Fig. 2 and Fig. 3a,c) display water masses typical of the Southern Boundary at the Greenwich Meridian as identified in previous studies (e.g., Orsi et al., 1995). Orsi et al. (1995) defined the Southern Boundary of the ACC as the location of the southernmost extent of UCDW, because it is the only water mass which is found exclusively in the ACC and not in the subpolar region to the south of the ACC. They identified UCDW as an oxygen-depleted water mass with temperatures greater than 1.5°C and practical salinities greater than 34.5. (Note that absolute salinity, which we use here, has values approximately 0.17 greater than practical salinity in this region). There is clearly an oxygen-depleted water mass at mid depths to the north which does not extend to the southern end of these transects (Fig. A1). Although the dissolved oxygen sensors on the gliders were calibrated by the manufacturers, no water samples were collected for in situ calibration of dissolved oxygen. There may therefore be some error in the absolute values due to the lack of in situ calibration, but we have no reason to doubt the relative values which show the oxygen-depleted layer. An offset was found between the two gliders, so we applied an intercalibration offset of +30 $\mu$mol kg$^{-1}$ to glider sg640 (transect B). Below approximately 250 m the salinities are $> 34.67$ throughout our data set, so salinity cannot be used to find the location of the Southern Boundary here. We therefore use the 1.5°C contour to determine the Southern Boundary's precise location. We find the Southern Boundary located between 55.55 and 55.82°S (spanning over 28 km), with the location for each transect shown as black dashed vertical lines on Fig. 2. This region displays greater variability (between transects) of temperature and salinity (Fig. 3) throughout the water column than surrounding areas to the north and south. The warming and freshening of the near surface water masses (top 50 m) from transect A to transect E (Fig. 2) are due to solar radiation and sea ice melt respectively as expected during austral spring and summer (October-January observations). The seasonal warming and freshening causes properties, such as temperature and salinity, to deviate from the mean sections more strongly near-surface than at depth (Fig. 3b,d). Antarctic Surface Water (AASW) occupies the top 150-200 m. To the north of the Southern Boundary, the AASW lies above Upper Circumpolar Deep Water (UCDW, 200-750 m), which in turn lies above Lower Circumpolar Deep Water (LCDW). To the south of the Southern Boundary, LCDW is found higher in the water column, below the AASW.

This highly variable frontal region is identified by greater standard deviations of temperature, salinity (Fig. 3b,d) and dis-
solved oxygen (Fig.A2) throughout the water column. UCDW extends furthest south in transect D, reaching 55.82°S (Fig. 2d,i), and least far south in transect C, reaching only 55.5°S (Fig. 2c,h). Although Orsi et al. (1995) did not consider the Southern Boundary to be associated with a circumpolar frontal jet, other authors have found the Southern Boundary to be co-located with a frontal jet at the Greenwich Meridian (Billany et al., 2010; Swart et al., 2010), and thus co-located with strong gradients in ADT. Similarly, our data shows strong eastward DACs and surface drift (Fig. 4), clear eastward jets in the geostrophic velocities (Fig. 5b,d) and strong ADT gradients (Fig. 5a,c), co-located with the Southern Boundary identified using the water mass-based definition of Orsi et al. (1995). Billany et al. (2010) further demonstrated that the Southern Boundary was co-located with the southernmost strong ADT gradients across the ACC at the Greenwich Meridian (using data collected between 1993 and 2008). We find the southernmost strong ADT gradients, shown as grey dashed lines on Fig. 5 approximately 8 to 30 km to the south of the locations found using the water mass-based definition of the Southern Boundary. Sokolov and Rintoul (2009a) suggested that the fronts of the ACC were associated with fixed SSH contours, but later authors(Gille, 2014; Gille et al., 2016) demonstrated that this was not the case over longer timescales, such as multiple years. However, since our dataset only spans two months we are not concerned with effects seen over multiple years. We have therefore identified the ADT contour, -1.16 m, which most closely matches the location of the strong ADT gradients associated with the frontal jet of the Southern Boundary, and use this to map daily positions of the frontal jet, as shown in Fig. 1.

Across the Southern Boundary, most transects (A,B,D,E) have strong horizontal density gradients (Fig. 2 and Fig. 3). The $27.73 \text{ kg m}^{-3}$ and $27.93 \text{ kg m}^{-3}$ isopyncals mark the upper and lower boundary of UCDW north of the Southern Boundary and are in general shallowing to the south. The $27.93 \text{ kg m}^{-3}$ isopycnal slopes strongly at the location of the Southern Boundary. In addition, in some areas, individual for each transect, the $27.73 \text{ kg m}^{-3}$ isopycnal bowls downward. These areas coincide with colder and fresher water mass properties than in the ambient water in the upper 250 m of the water column and are located 30-70 km south of the Southern Boundary. In contrast, transect C has weaker horizontal density gradients than the other transects, which is implied by a less steeply sloping $27.93 \text{ kg m}^{-3}$ isopycnal. The $27.73 \text{ kg m}^{-3}$ isopycnal in transect C also does not bowl downwards and does not show the changes in water mass properties, associated with the bowl-structure, seen in the other transects.

In summary, we have shown that the location of the Southern Boundary and its frontal structure change on short time scales (approx. 15.3 days, the average interval between the times when the glider was at the precise location of the Southern Boundary on successive transects). The following section will focus on transects A and C as a case study to identify the processes that influence the frontal structure and specifically modify the horizontal density gradients across the front. Comparable figures for the other transects are included in Appendix A for completeness.

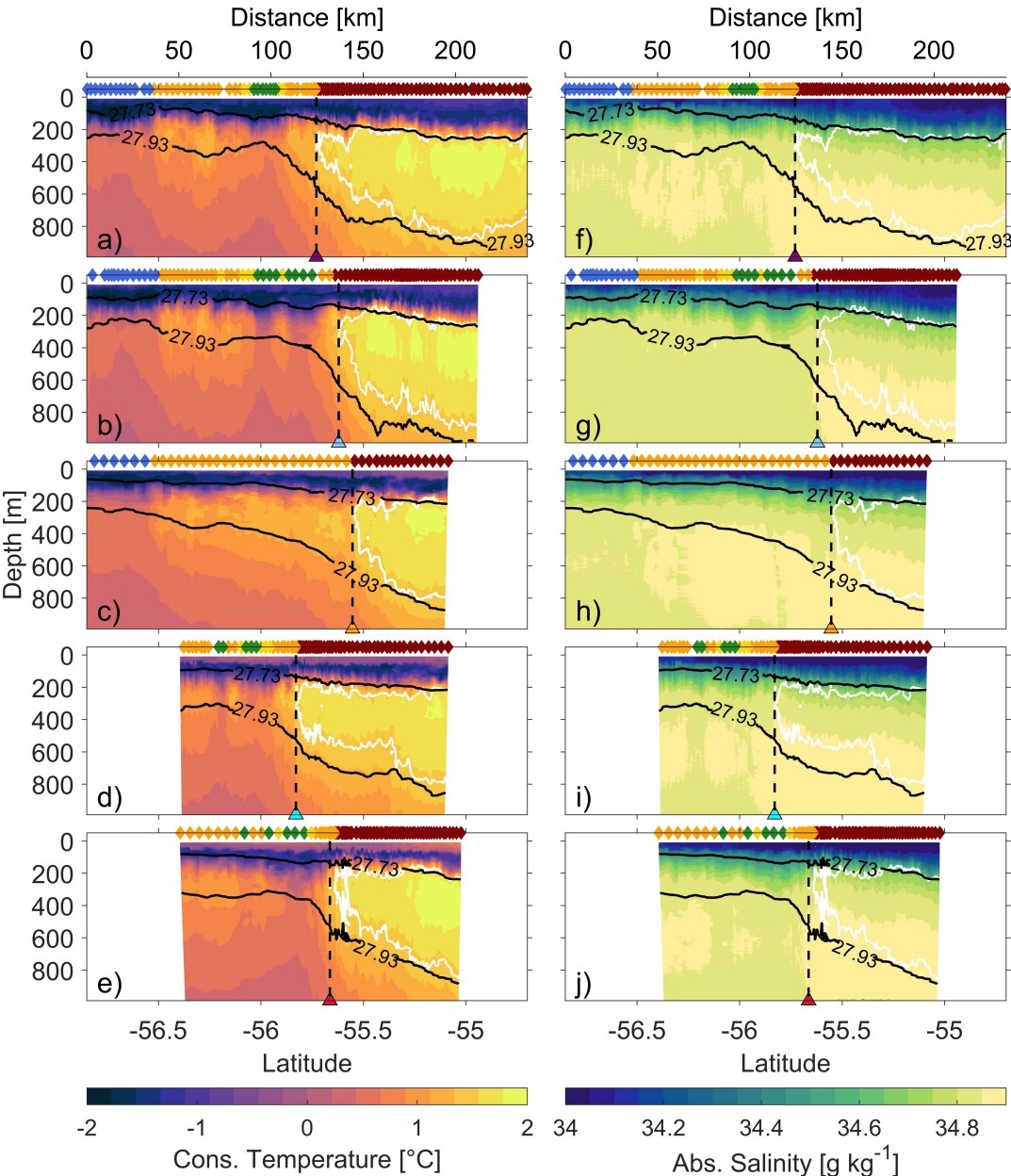

**Figure 2.** Hydrography of glider transects A-E showing conservative temperature (left column, panels a-e) and absolute salinity (right column, panels f-j) for each transect. Potential density contours of 27.73 kg m$^{-3}$ and 27.93 kg m$^{-3}$ are shown in black. The 1.5°C isotherm is shown with white contours. The triangles at the bottom of each panel, and the black dashed line extending upwards from each triangle, indicate the location of the Southern Boundary defined as the southernmost extent of UCDW (Orsi et al., 1995). The triangles are colored for each individual transect as in Fig. 1, and the same transect color coding is used in Fig. 3. The colors at the top of each panel represent our classification into areas north of the Southern Boundary (red), within a transition zone (orange), within the core of an eddy (green) and on the outer edges of an eddy (yellow), and south of the Southern Boundary (blue). This colour coding is discussed in section 3.

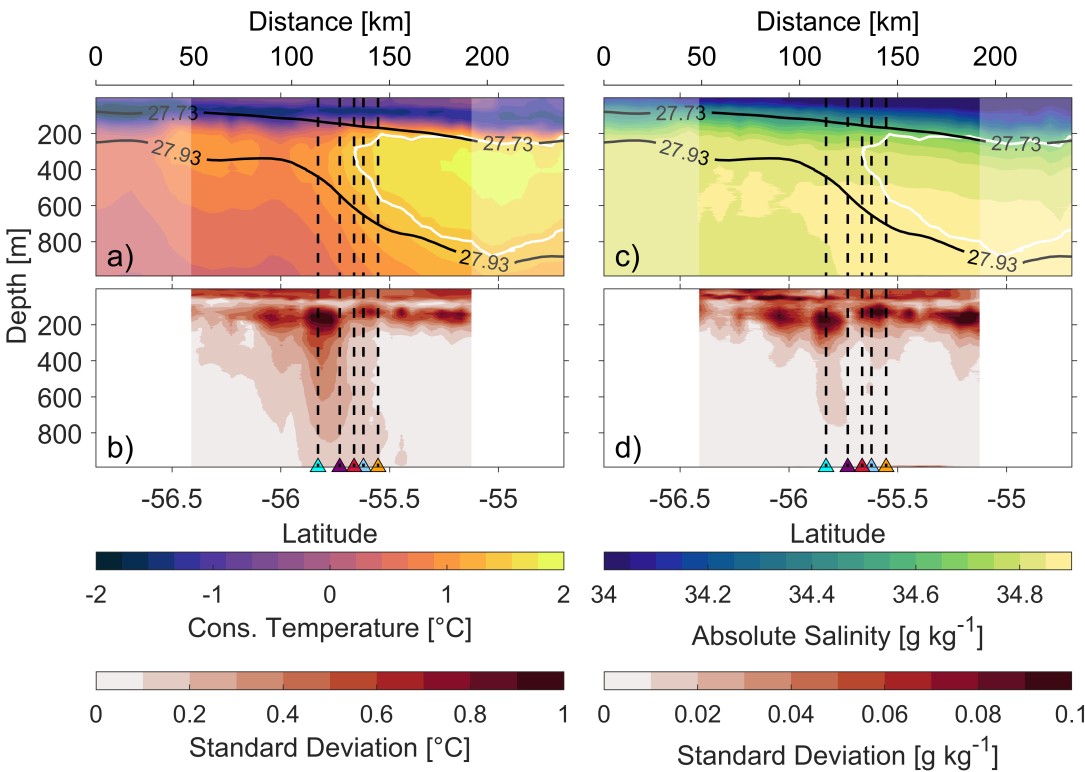

**Figure 3.** Mean (a,c) and standard deviation (b,d) of all glider transects A-E for (a,b) conservative temperature and (c,d) absolute salinity. The colored triangles at the bottom of panels (b,d), and black dashed lines extending upwards from them are as in Fig. 2. Data from each transect are binned to the same 5 km horizontal grid and then averaged (mean) for all transects. Partially shaded areas on (a,c) indicate areas that do not have data from all transects. Mean isopycnals 27.73 kg m$^{-3}$ and 27.93 kg m$^{-3}$ are shown in black. The white contour indicates the mean 1.5 °C contour.

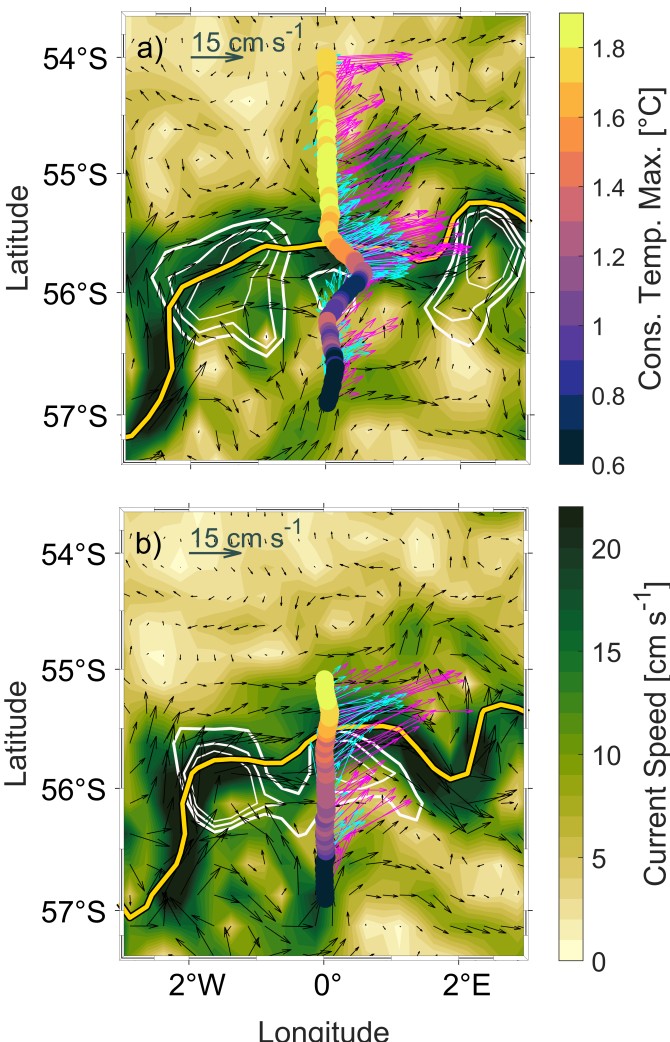

**Figure 4.** Maps of altimetric sea surface geostrophic velocities (velocity vectors in black and speed in green shading) during the glider crossings of the Southern Boundary for (a) $18^{th}$ November 2019 (transect A) and (b) $3^{rd}$ December 2019 (transect C). Bold yellow contours indicate the -1.16 m ADT contour on the same days. Coloured dots show the temperature maximum for each vertical profile along the respective transect. Glider dive average currents (cyan vectors) and glider surface drift speeds (magenta vectors) are superimposed. White contours show sea level anomalies of 0.06, 0.07 and 0.08 m (thin to bold, eddy core to eddy edge) and mark the location of clockwise eddies in close proximity to the Southern Boundary.

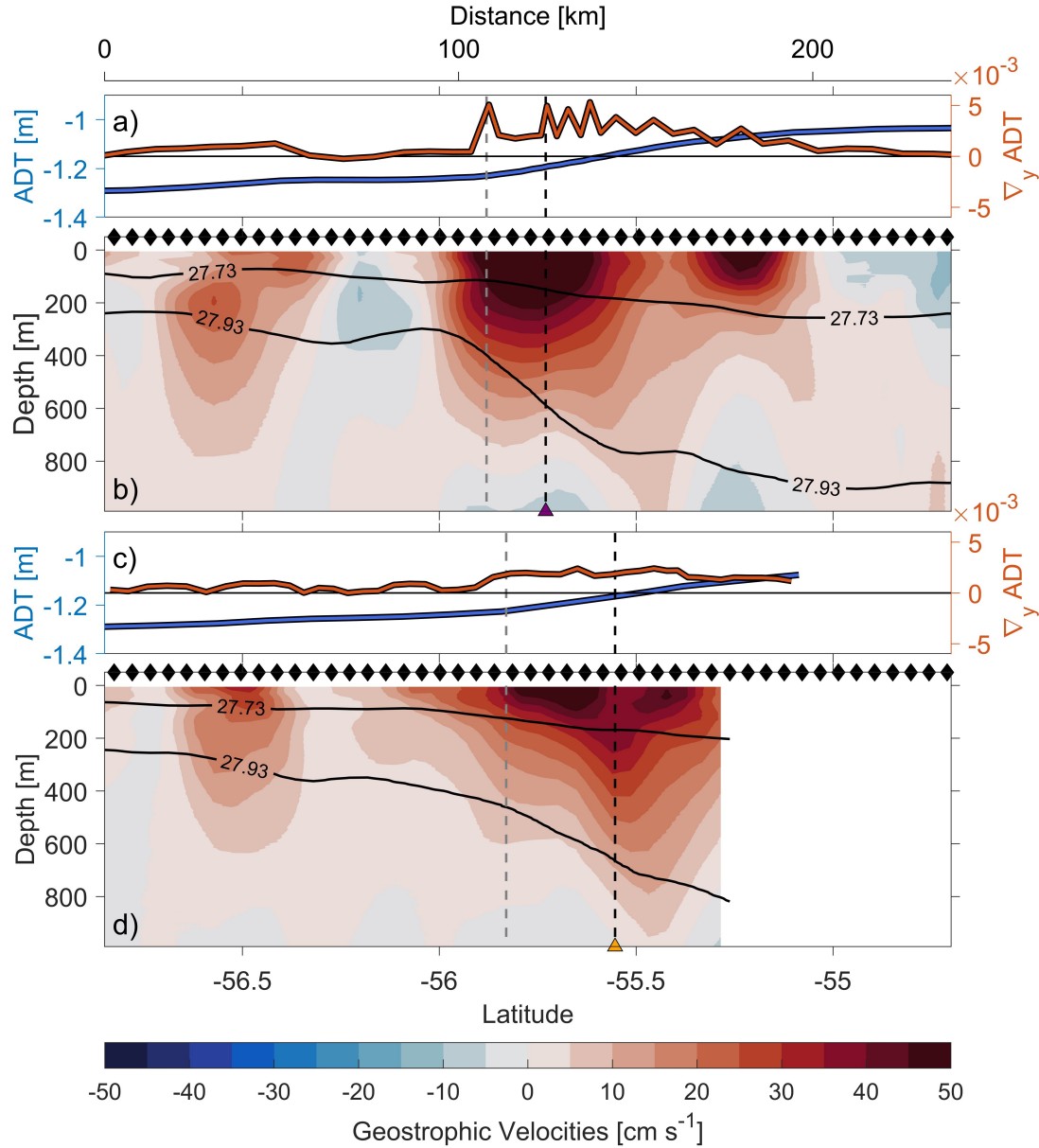

**Figure 5.** Real-time altimetric ADT and gradients of ADT ($\nabla_y \text{ADT}$) for (a) transect A and (c) transect C. (b,d) Geostrophic velocities perpendicular to the respective glider transects A and C and referenced to the DAC with a horizontal smoothing (moving mean filter) of approx. 15 km (Rossby radius within the region of interest). Positive geostrophic velocities are defined as eastward (red). Black contours are as in Fig. 2. The black diamonds at the top of each panel show the uniform grid spacing for both transects as described in section 3. The dashed grey lines indicate the location of the Southern Boundary's frontal jet based on the southernmost strong ADT gradient. The coloured triangles at the bottom of panels (b,d), and black dashed lines extending upwards from them are as in Fig. 2 and show the water mass-based location of the Southern Boundary.

## 3    Effects of Mesoscale Eddies on Frontal Structure and Frontal Jet

In this study, mesoscale eddies in close proximity to the Southern Boundary (between 54.88°S and 56.63°S) are identified using SLA as introduced in section 2. Three specific contours of SLA (0.06, 0.07 and 0.08 m) are chosen to identify mesoscale eddies influencing the frontal structure of the Southern Boundary (Fig. 4). For transect A, the SLA contours (Fig. 4a) reveal a mesoscale eddy (approx. 20-30 km wide) located to the south of the Southern Boundary at the Greenwich Meridian. In transect A, the glider captures the eddy's core and thus provides information on its rotational direction as well as its water mass properties. Surface current velocities from the altimetry, DAC and surface drift from the glider (Fig. 4a) provide evidence for a clockwise-rotating eddy with eastward velocities at its northern edge (55.8°S) and westward velocities at its southern edge (56.2°S). Due to its clockwise rotation, it is implied that the identified eddy is a cold-core eddy. Within a matter of days, the clockwise eddy is advected to the east with the ACC or merged with a larger structure such as the jet's meander to the west or east and is thus not captured again in transect C. As a result, neither sea surface slopes nor DAC nor surface drift from the glider (Fig. 4b) indicate a clockwise rotation during transect C. Thus, transect C contains information on the Southern Boundary's frontal structure without an eddy influencing it.

The temperature maximum for each vertical profile of transects A and C (Figs. 4 and 6) displays higher temperatures ($\Theta > 1.8°C$) north and lower temperatures ($\Theta < 0.65°C$) south of the Southern Boundary, thus demonstrating a high to low temperature gradient from north to south in both transects. The main differences between transect A and C in the temperature maximum occur in the transition zone between the warm regime in the north and the cold regime in the south. This is also the region where the clockwise eddy, captured in transect A, is located. Within the transition zone, the temperature maximum in transect A demonstrates significantly lower temperatures (0.65-0.8°C at 56°S) than in transect C (Fig. 6), which provides further evidence that the captured eddy in transect A is a cold-core eddy.

The temperature of the temperature maximum for each transect further is used to divide each transect into segments (Fig. 6). We introduce a colour-coding scheme to show whether a vertical profile is located to the north or south of the Southern Boundary, within the transition zone or within the cold-core eddy. The colour-coded segments are used in Figs. 2, 6, 7, 8, 5, 9 and 10. Each transect is segmented by the following criteria:

**Table 1.** Segmentation values based on the temperature of the temperature maximum for each vertical profile across each transect. The colour-coding scheme introduced in Fig. 6 is based on these values.

| Segments | $\Theta_{max}$ [°C] |
|---|---|
| North of the Southern Boundary | > 1.5 |
| Temperature Transition Zone | 0.95-1.5 |
| Outer Eddy | 0.8-0.95 |
| Eddy Core | 0.65-0.8 |
| South of the Southern Boundary | < 0.65 |

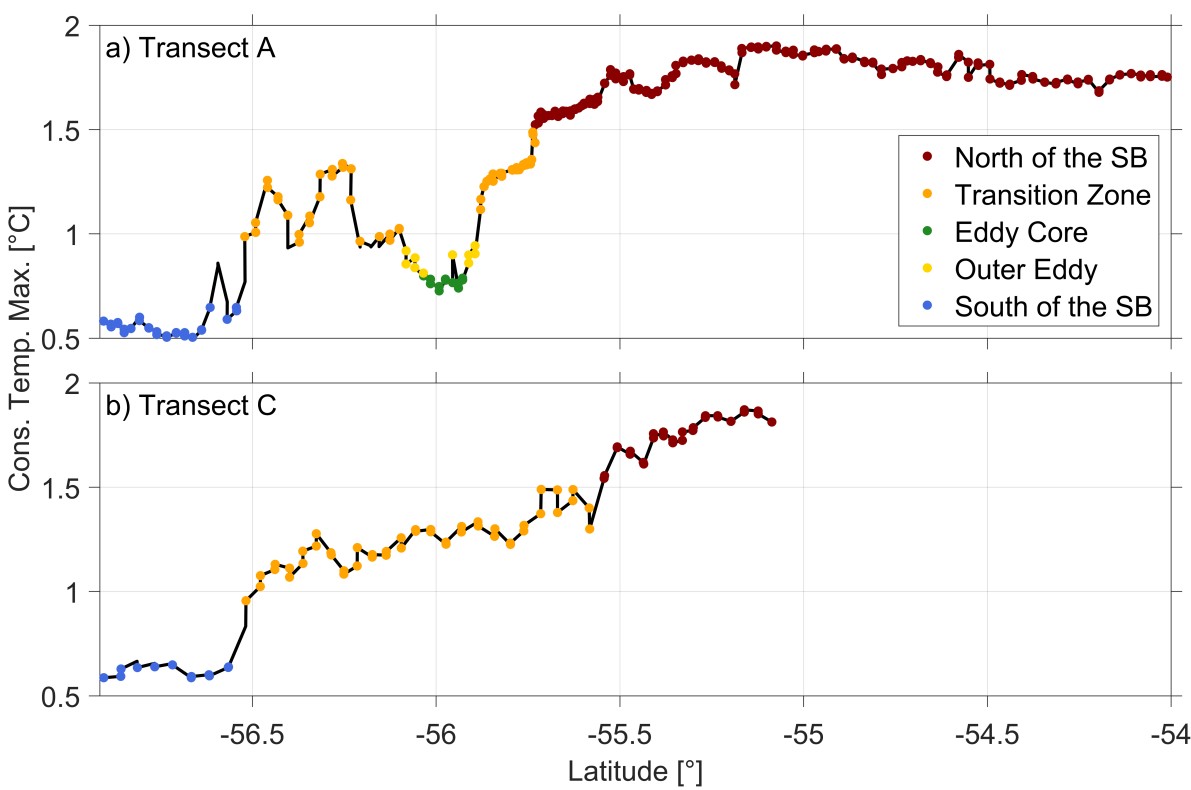

**Figure 6.** The color-coded temperature maximum for each vertical profile along (a) transect A and (b) transect C. The segmentation values for each regime, on which the colour-coding is based, are defined in Table 1 in section 3 .

North of the Southern Boundary, Θ/S profiles (Figs. 7 and 8, black arrows) converge towards similar temperatures and salinities (Θ > 1.5°C and S>34.7 g kg$^{-1}$)) at the base of the thermocline (230 m), which represents UCDW. LCDW is identified below the UCDW layer with slightly lower temperatures (Θ ≈ 0.6°C) and similar salinities (S>34.7 g kg$^{-1}$). In the temperature transition zone, lower temperatures between 0.2-1.2°C indicate the presence of moderately to heavily modified CDW (mCDW), which is representative of the temperature transition zone in both transects. The clockwise eddy identified in transect A (Fig. 7 a,d,e) presents properties similar to the cold regime but with slightly higher temperatures (about 0.4 to 0.6°C higher) below the thermocline and slightly reduced salinities above the depth of the thermocline. The similar water mass properties of the eddy and the cold regime suggest that the eddy originated south of the Southern Boundary.

Geostrophic velocities referenced to the DAC (Fig. 5) reveal the frontal jet associated with strong density gradients. Tran-

sect A has the most intense jet with an eastward core velocity of up to 80 $\text{cm s}^{-1}$ and a meridional extent of about 50 km. In contrast, the frontal jet associated with smaller density gradients is weaker and broader in transect C with an eastward core velocity of up to 60 $\text{cm s}^{-1}$ and a meridional extent of about 80 km.

The southern edge of the frontal jet is located between 55.5-56°S in both transects and is consistent with the Southern Boundary's location indicated by both the southernmost limit of UCDW (Fig. 2) and the gradient of altimetry-derived ADT (Fig. 5a,c). Westward velocities south of the Southern Boundary at a depth of 80-400 m between 56-56.5°S in transect A are consistent with the location and cyclonic rotation of the eddy (Fig. 4a) and indicate the eddy's southern edge. Both transects (Fig. 5a,b) show another eastward flow further south (56.25-56.75°S) with velocities of up to 30 $\text{cm s}^{-1}$, which marks the boundary between the transition zone and the cold regime south of the Southern Boundary.

In summary, we have shown that a clockwise, cold-core eddy located south of the Southern Boundary influences the frontal structure and strengthens the density gradients across the front and strengthens its frontal jet. Additionally, we have demonstrated that density gradients and the frontal jet are weaker across the Southern Boundary after the eddy has been advected eastward. Furthermore, the characterization of different regimes across the Southern Boundary region using the temperature maximum for each vertical profile reveals specific water mass properties for each regime. We further established, based on the similar water mass properties of the cold-core eddy and the cold regime, that the eddy originated south of the Southern Boundary. In the following section, we address how the changes in the frontal structure of the Southern Boundary impact its barrier/blender properties.

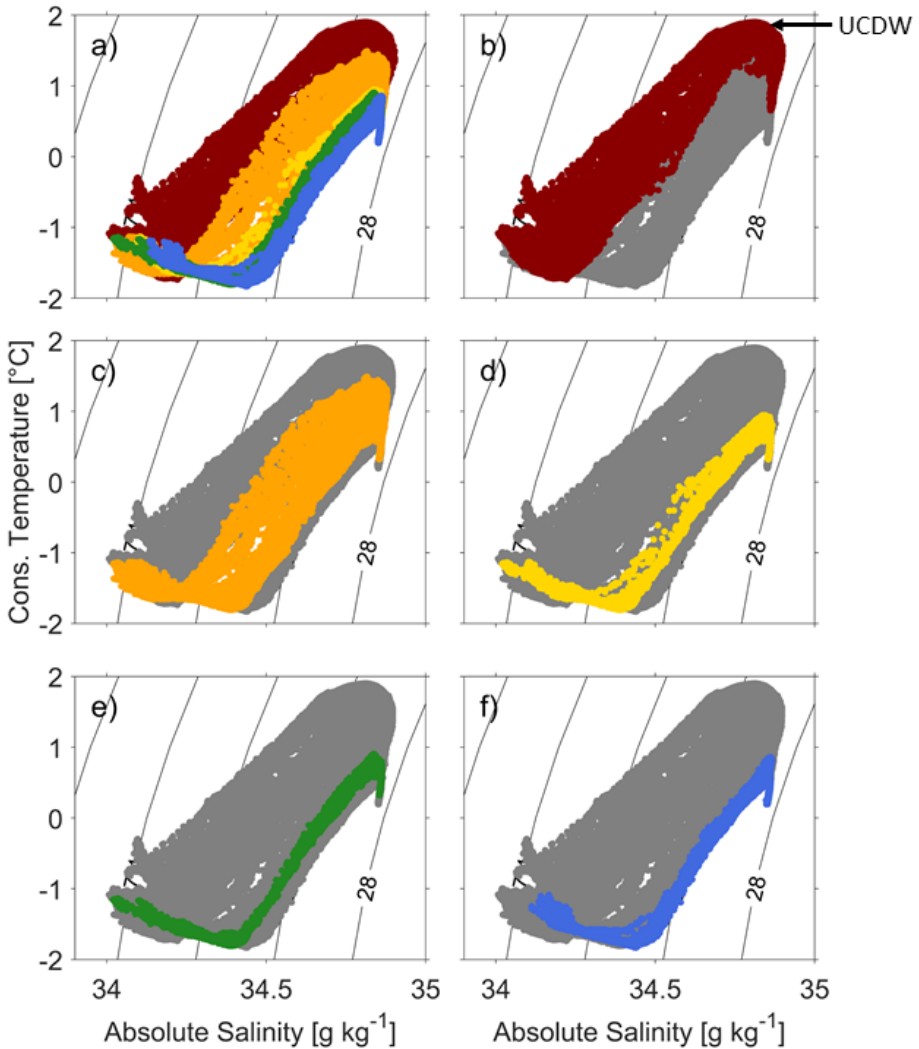

**Figure 7.** Θ/S diagrams for Transect A with colour-coding defined in Table 1 and colours shown in Fig. 6. (a) All profiles of transect A shown with colour-coding. (b-f) Grey dots show all profiles in the transect, with coloured dots showing the profiles: (b) north of the Southern Boundary, c) in the temperature transition zone, d) in the outer eddy, e) in the eddy core and f) south of the Southern Boundary.

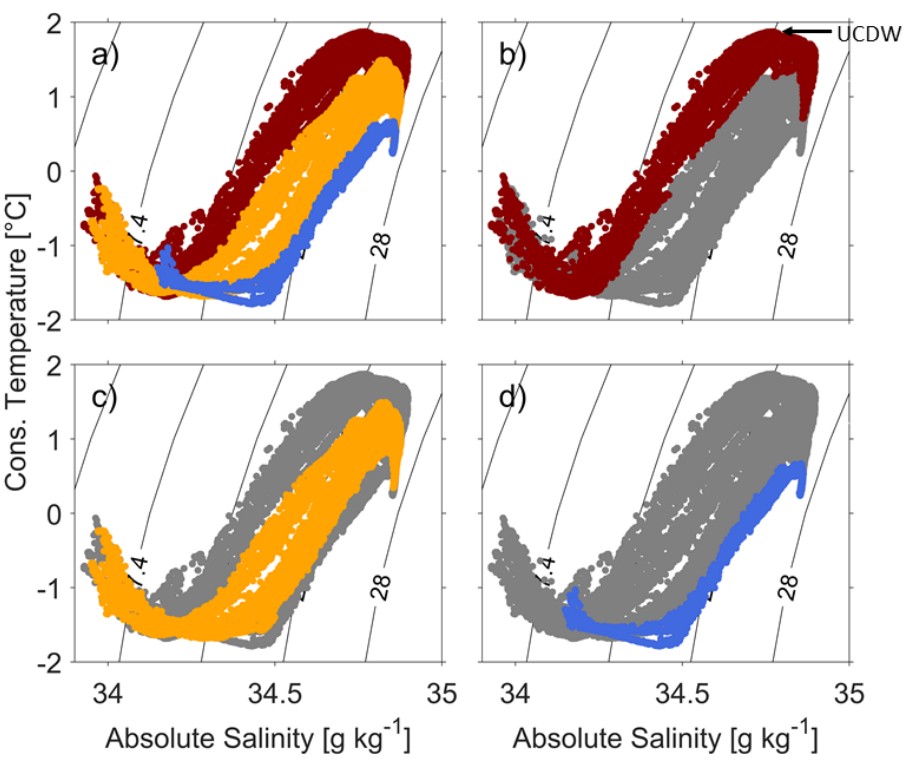

**Figure 8.** As for Fig. 7 but for transect C.

## 4 Effects of Mesoscale Eddies on Mixing Length Scales

### 4.1 Mixing Length Scale Diagnostics

Ferrari and Nikurashin (2010) demonstrated that strong mean flows, such as those found within the ACC fronts, can suppress lateral mixing. Naveira-Garabato et al. (2011) further quantified this effect by estimating mixing length scales (MLS) across the three major fronts of the ACC using hydrographic sections with a large scale resolution (approx. 50 km station separation). Naveira-Garabato et al. (2011) found that the eddy diffusivities are typically suppressed across the ACC's frontal jets, primarily as a result of reduced mixing lengths. Here we calculate MLS from the highly-resolved transects A and C to assess whether the passage of an eddy across the Greenwich Meridian affects the ability of water to mix across the Southern Boundary. The applied method slightly differs from the method used by Naveira-Garabato et al. (2011) as their study used ship-based hydrographic sections rather than closely-spaced glider transects. This study is thus based on a method for glider data described by Dove et al. (2023); Viglione (2019). All data used for the calculations are on a uniform 5 km grid as stated in section 3.

First, the 5 km x 2 m gridded temperature is linearly interpolated in the vertical onto a potential density grid with an interval of $0.02 \, \text{kg m}^{-3}$, as used by Naveira-Garabato et al. (2011). We will refer to this field as $\Theta_\rho$. Next, a large-scale temperature field, $\Theta_m$, and a large-scale gradient along potential density surfaces, $\nabla_\rho \Theta_m$ are generated by spatially smoothing with a 30 km (twice the Rossby Radius) x $0.08 \, \text{kg m}^{-3}$ moving median filter to filter out small scale effects. Finally, at each grid point we find the root mean square difference $\Theta_{rms}$ between the value of $\Theta_m$ at that grid point, and the values of $\Theta_\rho$ within a 5-element window in the horizontal (i.e., on the same density surface) centered on that grid point. In other words,

$$\Theta_{rms,i} = \sqrt{\frac{\sum_{j=i-2}^{j=i+2}(\Theta_{\rho,j} - \Theta_{m,i})^2}{5}} \tag{1}$$

where $i$ is an index from south to north along a potential density surface. The mixing length scales, $L_{mix}$, are then calculated from:

$$L_{mix} = \frac{\Theta_{rms}}{\nabla_\rho \Theta_m}. \tag{2}$$

We further calculate potential vorticity, which is a largely and materially conserved property in the ocean interior that can be used to identify the susceptibility of the flow to instabilities (Haine and Marshall, 1998). The Ertel potential vorticity (PV) can be written as:

$$Q = (f\hat{k} + \nabla \times \mathbf{u}) \cdot \nabla b, \tag{3}$$

where f is the Coriolis parameter, g the gravitational acceleration, $\nabla \times \mathbf{u}$ is the relative vorticity, where $\mathbf{u}$ the velocity vector. $b = -g((\rho - \rho_0)/\rho_0)$ is the buoyancy, where $\rho$ is the ocean density and $\rho_0$ is the reference ocean density. In this study, the glider transects only provide the cross-section (along-stream) velocity component. Therefore the PV has to be simplified to achieve the observational PV (Azaneu et al., 2017).

$$PV = -\frac{\partial v}{\partial x}\frac{\partial b}{\partial v} + \frac{\partial v}{\partial x}\frac{\partial b}{\partial z} + f\frac{\partial b}{\partial z}, \tag{4}$$

where the first and second term correspond to the horizontal and vertical components of the relative vorticity. The third term corresponds to the stretching term which is proportional to the vertical stratification (Azaneu et al., 2017). The observational PV simplification assumes that the along-stream buoyancy gradients are much weaker than the cross-stream buoyancy gradients. The PV is mapped on potential density surfaces with the same vertical and horizontal gridding as for the calculation of
$L_{mix}$.

The MLS diagnostics show differences between transect A (Fig. 9) and transect C (Fig. 10). The magnitude of $\nabla_\rho \Theta_m$ across the Southern Boundary is substantially larger in transect A (Fig. 9c) than transect C (Fig. 9c), where maximum $\nabla_\rho \Theta_m$ aligns with the southern edge of the frontal jet (Fig. 5) in both transects. The transects further demonstrate enhanced $\nabla_\rho \Theta_m$ near
56.5°S representing the second, weaker velocity core that marks the boundary between the transition zone and the cold regime further south. The magnitude of $\Theta_{rms}$ in both transects (Fig. 9b and Fig. 10b) is largest along the upper boundary of UCDW and mCDW and reflects fluctuations in the transition to denser water masses below and to the south. The resulting $L_{mix}$ for transect A (Fig. 9d) and transect C (Fig. 10d) differ by an order of magnitude across the Southern Boundary. Transect A exhibits near zero $L_{mix}$ distinctly confined between 55.5-56°S within the 27.6 $\mathrm{kg\,m^{-3}}$ and 28 $\mathrm{kg\,m^{-3}}$ isopycnals. The region
of low $L_{mix}$ coincides with the region of strongest eastward velocities of the frontal jet and southernmost gradients of ADT (Fig. 5a,b) as well as strongest magnitudes of $\nabla_\rho \Theta_m$ (Fig. 9c). The low magnitude of $L_{mix}$ in transect A is classified as eddy suppressing as values as eddy diffusivities are proportional to $L_{mix}$ ($\kappa = U_e L_{mix} c_e$, where $U_e$ is the eddy velocity scale, and $c_e$ is the eddy mixing efficiency). These results imply that the ability of water to move across the Southern Boundary is suppressed for transect A, which is consistent with all other transects that have strong density gradients across the Southern
Boundary (transects B, D and E; Appendix B Figs. B1, B2 and B3). The stronger density gradients (transects A, B, D and E) are associated with eddies passing the Greenwich Meridian south of the Southern Boundary and influencing its frontal structure and frontal jet. In contrast, transect C exhibits increased $L_{mix}$ of up to 20 km across the Southern Boundary, and the region of lower $L_{mix}$ is not as clearly confined to the region between 55.5-56°S as in transect A, which suggests that the ability of water to flow across the Southern Boundary is increased in transect C.


In general, PV is largest near the surface and decreases with depth towards zero (Figs. 9e and 10e). Between the 27.5 $\mathrm{kg\,m^{-3}}$ and the 27.7 $\mathrm{kg\,m^{-3}}$ isopycnals in transect A, the PV increases sharply from PV$<< -8 \cdot 10^{-9} s^{-3}$ (south of the Southern Boundary) to PV$\approx -3 \cdot 10^{-9} s^{-3}$ (north of the Southern Boundary). The mesoscale eddy influencing the Southern
Boundary leads to larger PV (centred around 56°S) coinciding with the eddy's location determined from the temperature maximum (colour-coding). In contrast, the PV in transect C does not increase sharply between the 27.5 $\mathrm{kg\,m^{-3}}$ and the 27.7 $\mathrm{kg\,m^{-3}}$ isopycnals. In transect A, at the 27.8 $\mathrm{kg\,m^{-3}}$ isopycnal, which marks the upper boundary of UCDW (centered within the isopycnals of low $L_{mix}$), the PV also increases sharply across the front from PV$\approx -2.59 \cdot 10^{-9} s^{-3}$ (south of the Southern Boundary) to PV$\approx -0.8 \cdot 10^{-9} s^{-3}$ (north of the Southern Boundary) in a more pronounced way than in transect C. Similar to
the analysis of (Bower et al., 1985) for the Gulf Stream, the gradients in PV across the Southern Boundary indicate enhanced

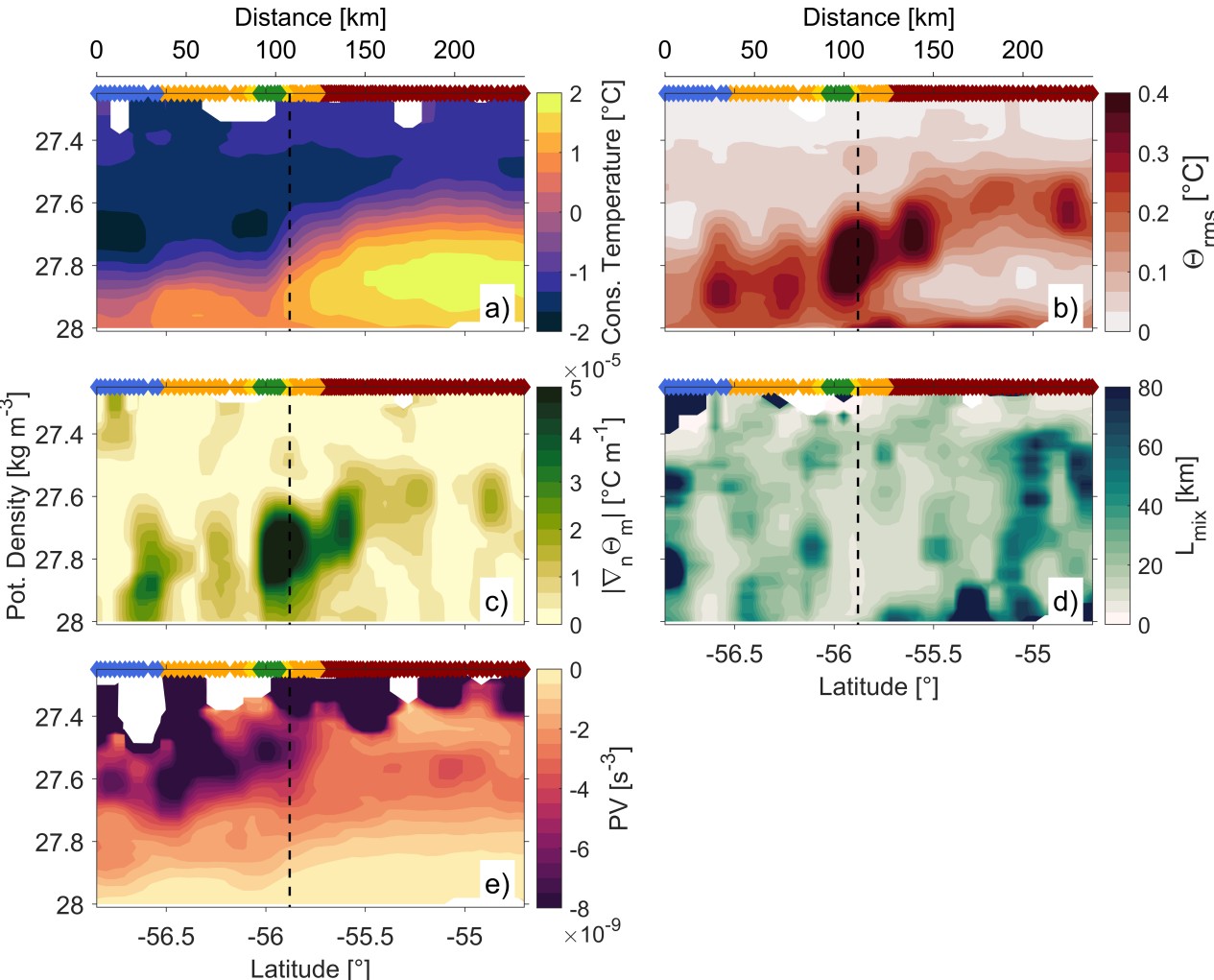

**Figure 9.** (a) The mean temperature field $\Theta_m$, (b) the measure of the temperature fluctuations $\Theta_{rms}$, (c) the gradient of the mean temperature $\nabla_\rho \Theta_m$ along potential density surfaces, (d) the mixing length scales $L_{mix}$ and (e) the potential vorticity (PV) for transect A. All panels are spatially smoothed by a 30 km x 0.08 kg m$^{-3}$ moving median filter. The dashed black line indicates the location of the Southern Boundary as defined with gradients of ADT as shown in Fig. 5. All subfigures a-e are shown in density space with a vertical gridding of 0.02 kg m$^{-3}$. The colour-coded diamonds at the top of each panel describe the segments along transect A as defined in Table 1

barrier-like properties in transect A compared with transect C. Although the PV gradients and low $L_{mix}$ indicate an impedance to cross-frontal mixing, increasing $L_{mix}$ values near the surface suggest that there is still some exchange taking place between regions north and south of the front.

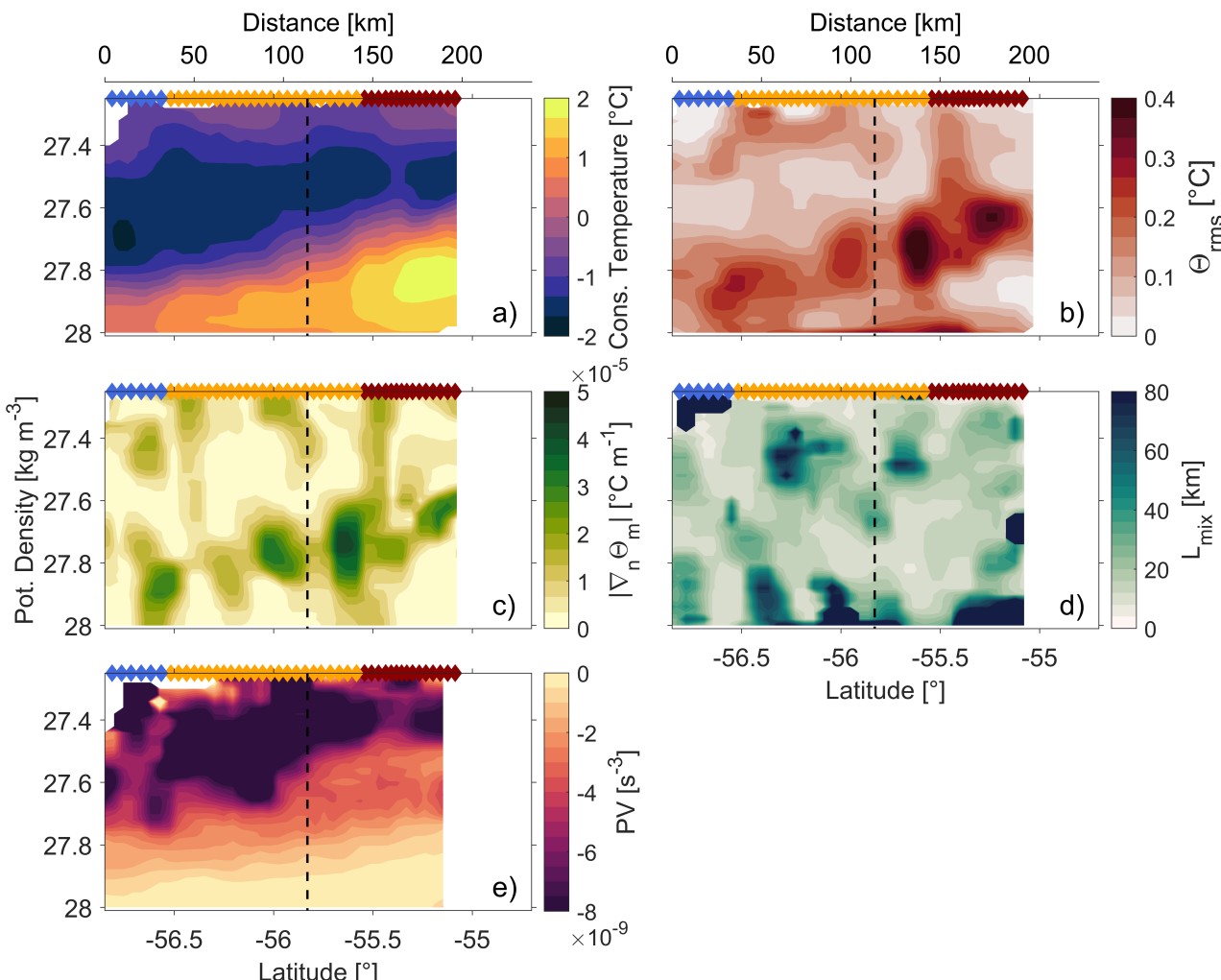

**Figure 10.** As for Fig. 9 but for transect C.

These results indicate that changes in $L_{mix}$ and differing PV gradients in transects A and C are linked to the mesoscale cold-core eddy influencing the density gradients across the Southern Boundary in transect A. As revealed in section 3, the cold-core eddy passing the Greenwich Meridian interrupts the temperature transition zone, strengthens the density gradients across the Southern Boundary and amplifies the frontal jet in transect A. The suppressed $L_{mix}$ and more pronounced PV gradients imply that the ability of properties such as heat and freshwater to cross the Southern Boundary is dampened. In contrast, after the cold-core eddy has been advected away to the east, we find no interruption of the temperature transition zone, weakened density gradients and a broader and weakened frontal jet with increased and less confined $L_{mix}$ and less pronounced PV

gradients across the front which further suggests the increased ability to exchange water mass properties across the Southern Boundary.

## 4.2 Interannual Variability of Barrier Properties

We have shown that the Southern Boundary's barrier properties (as represented by MLS) are related to the magnitude of the frontal jet. Therefore the strength of the frontal jet, routinely monitored by satellite altimetry, can be used as a proxy to determine the variability of the barrier strength of the Southern Boundary over long time scales (28 years of altimetry data are available). Previous studies have concluded that the use of a fixed ADT contour to define the location of the ACC's fronts is inappropriate over long time scales (such as multiple years), since even if seasonal cycles were removed the long-term warm-

ing of ACC waters and associated changes in ADT would not be eliminated (thermal expansion, (Gille, 2014)). Therefore, we estimate the Southern Boundary's location and barrier properties from 1993 to 2020 using surface frontal jet speeds calculated from sea surface slopes (Fig. 11) rather than using specific contours of ADT. The chosen speed contour to highlight enhanced frontal jet speed is 14.5 [$\mathrm{cm\,s^{-1}}$], which is determined from the mean frontal jet speed averaged across the latitude band from 54.87-56.62°S plus twice the standard deviation.


The Southern Boundary's location (determined from the frontal jet) has not migrated south and remains within the 54.87-56.62°S latitude band throughout the 1993-2020 record (Fig. 11). This contrasts with the reported southward migrating and intensifying westerly winds over the same time period (e.g., Chapman et al., 2020; Gille, 2014; Graham et al., 2012). The average frontal jet speeds across the latitude band (Fig. 11d) indicate that the frontal jet speed has accelerated over the past

decade (>14.5 $\mathrm{cm\,s^{-1}}$ between 2012-2020). Note that different values between geostrophic velocities and surface velocities derived from altimetry data are to be expected as satellite altimetry-derived currents are necessarily temporally and spatially smoothed by the process of creating the gridded product from relatively widely-spaced altimetric tracks infrequently repeated. This may lead to eddies and fronts being in the correct location, but smoothed in, for example, current speed, so that values from satellite altimetry tend to be smaller than observed current speeds. The increase in frontal jet speeds over the past decade

is associated with stronger gradients in ADT across the Southern Boundary that are indicated by an increase in ADT north of the Southern Boundary (Fig. 11). These results are consistent with Stewart (2021) and Shi et al. (2021) who demonstrated that the core eastward flow of the ACC has accelerated over the past decade. Stewart (2021) and Shi et al. (2021) further showed that the acceleration in eastward flow is related to the amplification of meridional density gradients in response to upper ocean warming within the ACC, rather than intensifying westerly winds. These results suggest that barrier properties of the Southern

Boundary have strengthened over the past decade, associated with strong gradients in PV and density leading to suppressed MLS as shown in section 4. The continuation of upper ocean warming may further increase meridional ADT gradients resulting in an accelerated frontal jet and strengthened barrier properties of the Southern Boundary in the future.

Satellite altimetry observations and eddy resolving models suggest an intensifying eddy field in response to stronger winds

(e.g., Meredith and Hogg, 2006; Hogg et al., 2015; Patara et al., 2016). At the Greenwich Meridian we see an increasing num-

ber of anticyclonic eddies (Fig. 11b) (and increased eddy kinetic energy, not shown) with increased current speeds south of the Southern Boundary (Fig. 11c) from 2009 onwards. Anticyclonic eddies moving south across the front may thus provide a more dominant contribution to the transport of properties across the Southern Boundary in the future as barrier properties strengthen through acceleration.

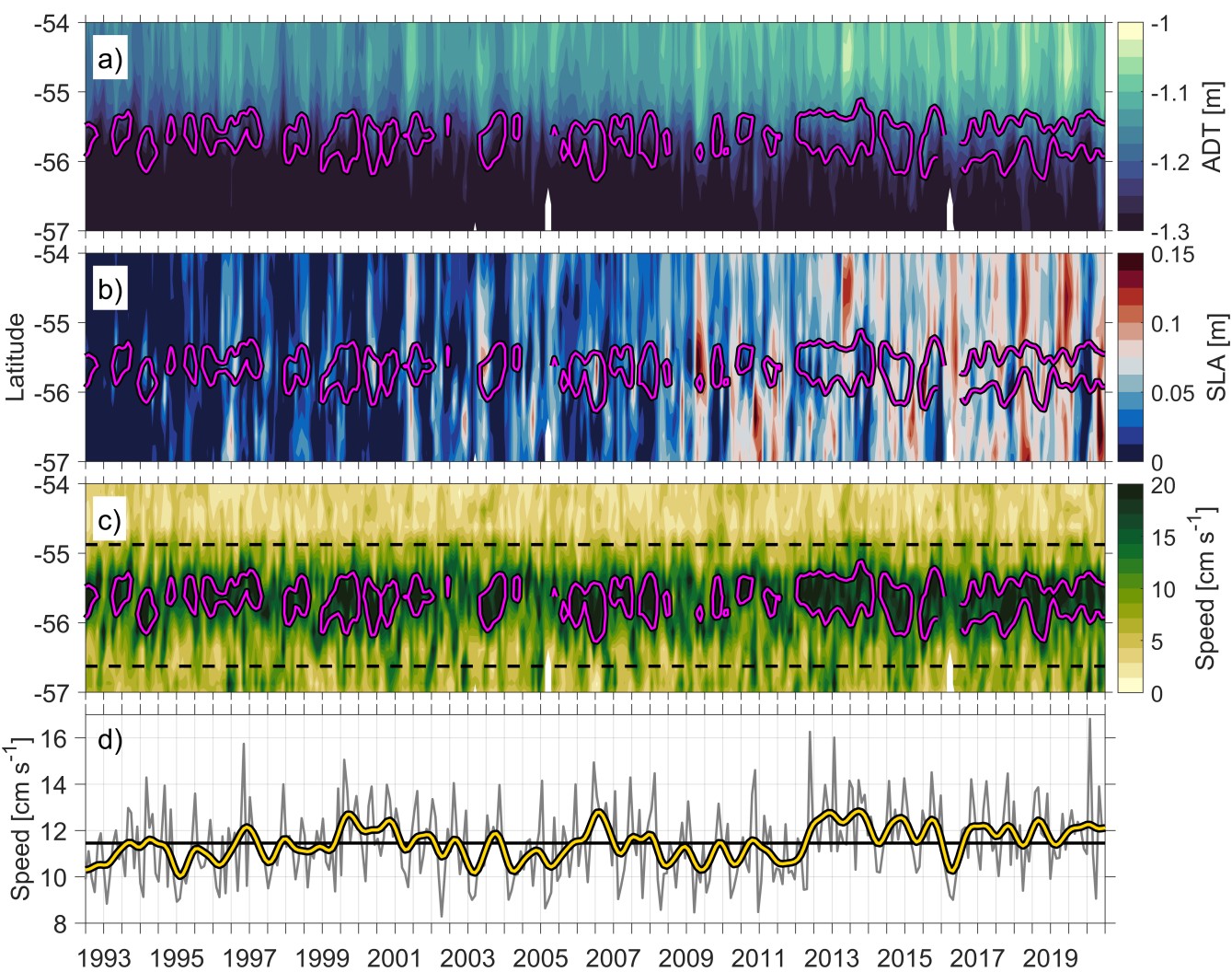

**Figure 11.** Hovmöller diagrams of a meridional transect at the Greenwich Meridian from altimetry showing (a) ADT, (b) SLA and (c) current speed. Panel (d) displays the averaged frontal jet speed within the 54.87-56.62°S latitude band (dashed black lines in (c)), with the monthly-average frontal jet speeds (grey) and smoothed (12-month moving median filter) frontal jet speeds (yellow). The horizontal black line in (d) is the mean frontal jet speed over the entire 28 years. The magenta contours in (a), (b) and (c) highlight periods of enhanced frontal jet speeds (>14.5 cm s$^{-1}$).

## 340  5  Conclusions

In this study we use two months of repeat, high-resolution glider transects over the Antarctic Circumpolar Current's Southern Boundary to assess its variability in location and intensity in terms of lateral gradients and velocities. During the observational time period, the Southern Boundary was located between 55.5-56°S, where the gradients in ADT and the southernmost limit of UCDW coincided. The estimated location is consistent with previously estimated locations of the Southern Boundary (e.g.,
Billany et al., 2010). Most glider transects (except transect C) are characterised by strong density gradients across the front associated with a strong frontal jet ($\approx 80 \mathrm{cm\,s^{-1}}$), whereas transect C demonstrated weaker density gradients associated with a weaker, broader frontal jet ($\approx 60 \mathrm{cm\,s^{-1}}$).

The glider transects and SLA revealed that mesoscale cold-core eddies influence the Southern Boundary's frontal structure
by disrupting the temperature transition zone, enforcing stronger density gradients across the front and amplifying the frontal jet. These findings are consistent with Williams et al. (2007) who demonstrated that eddies impact lateral density gradients across the ACC that can accelerate or decelerate the mean flow. We find that cold-core eddies are present in all transects that have a disrupted transition zone and strong density gradients (example transect A). In contrast, we show that the cold-core eddy in transect A is advected away eastward before transect C is occupied, which then does not cross the cold-core eddy and
presents weaker density gradients and a weaker, broader frontal jet. The highly energetic eddy field within the ACC varies rapidly and locally around the Antarctic continent and thus more observations are needed to address the impacts of mesoscale eddies on the Southern Boundary's frontal structure in a circumpolar fashion. Future investigations with a focus on regions where the influences of mesoscale eddies on the Southern Boundary's frontal structure are more or less significant may improve estimations of its barrier properties.


Low $L_{mix}$ and more pronounced PV gradients at the Southern Boundary are found at the upper boundary of UCDW (example transect A, Fig. 9 d) associated with strong density gradients and an amplified frontal jet. These characteristics suggest that the exchange of properties across the Southern Boundary is dampened. Thus, it is implied that strengthened barrier properties are a result of cold-core eddies enforcing stronger density gradients across the Southern Boundary. In contrast, increased
values of $L_{mix}$ of up to 20 km and less pronounced PV gradients across the Southern Boundary (example transect C, Fig. 10 d) are found when no cold-core eddy is observed to influence the Southern Boundary's frontal structure (weaker density gradients, weaker frontal jet). These findings emphasise that locally changing mesoscale structures can significantly impact the Southern Boundary's barrier properties and thus modulate cross-frontal exchange of properties such as heat and carbon. The cross-frontal exchange of properties is specifically relevant to regions where the Southern Boundary is located near the
continental shelf break, such as in the West Antarctic sector (e.g. Thompson et al., 2020).

Additionally, we have shown the importance of the linkage between MLS and the intensity of the frontal jet of the Southern Boundary to establish the Southern Boundary's barrier properties over long time periods (multiple years). Increased ADT

gradients across the Southern Boundary, as a result of increased ADT north of the front (Fig. 11a), confirm amplified frontal jet speeds from 2009 onwards. This is consistent with the amplification of the ACC's eastward flow demonstrated by Shi et al. (2021) and Stewart (2021). Based on our results, strengthened density gradients across the Southern Boundary and thus an intensified frontal jet are indicators for dampened cross-frontal exchange and strengthened barrier properties. Thus, we suggest that the poleward heat transfer through mixing across the Southern Boundary at the Greenwich Meridian has likely decreased over the last decade, whereas the intensified eddy field and generation of warm-core eddies that cross the Southern Boundary in response to intensifying westerly winds likely provided an increased contribution to poleward heat transfer. As these processes vary locally and temporarily, our results demonstrate that more investigations of the Southern Boundary's frontal jet intensity and barrier properties are needed to understand and estimate the cross-frontal exchange around the Antarctic continent in more detail.

*Data availability.* Quality controlled, vertically gridded data from the two Seagliders are available from the following links: SG537, DOI: 10.5281/zenodo.7472263 and SG640, DOI: 10.5281/zenodo.7472428. The altimetry and sea ice concentration data are both provided by the Copernicus Environment Monitoring Service (CMEMS) and are available at: DOI: https://doi.org/10.48670/moi-00148 (altimetry) and DOI: https://doi.org/10.48670/moi-00168 (OSTIA, sea ice concentration).

## Appendix A

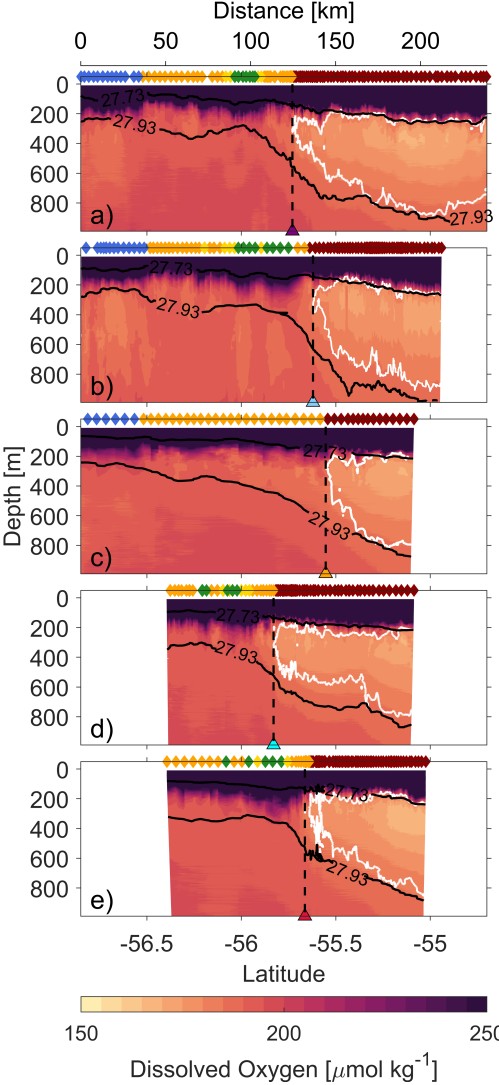

**Figure A1.** Dissolved oxygen for glider transects A-E. Although the dissolved oxygen sensors on the gliders were calibrated by the manufacturers, no water samples were collected for in situ calibration of dissolved oxygen. An offset was found between the two gliders, so we applied an offset to intercalibrate them, adding 30 micromoles per kg to glider sg640 (transect B). Potential density contours of 27.73 kg m$^{-3}$ and 27.93 kg m$^{-3}$ are shown in black. The 1.5°C isotherm is shown in white. The triangles at the bottom of each panel, and the black dashed line extending upwards from each triangle, indicate the location of the Southern Boundary defined as the southernmost extent of UCDW (Orsi et al., 1995). The triangles are coloured for each individual transect as in Fig. 1, and the same transect colour coding is used in Fig. 3. The colours at the top of each panel represent our classification into areas north of the Southern Boundary (red), within a transition zone (orange), within the core of an eddy (green) and on the outer edges of an eddy (yellow), and south of the Southern Boundary (blue). This colour coding is discussed in section 3.

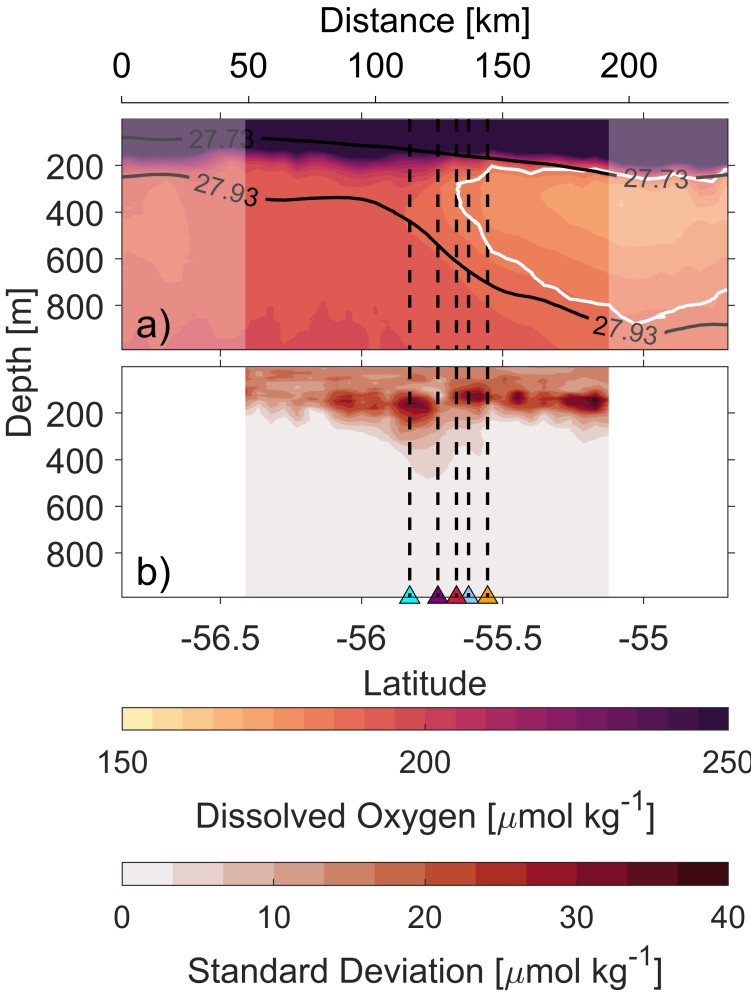

**Figure A2.** Mean (a) and standard deviation (b) of all glider transects A-E for dissolved oxygen. The coloured triangles at the bottom of panel (b), and black dashed lines extending upwards from them are as in Fig. 2. Data from each transect are binned to the same 5 km horizontal grid and then averaged (mean) for all transects. Partially shaded areas on (a) indicate areas that do not have data from all transects. Mean isopycnals 27.73 kg m$^{-3}$ and 27.93 kg m$^{-3}$ are shown in black. The mean 1.5 °C isotherm is shown in white.

**Appendix B**

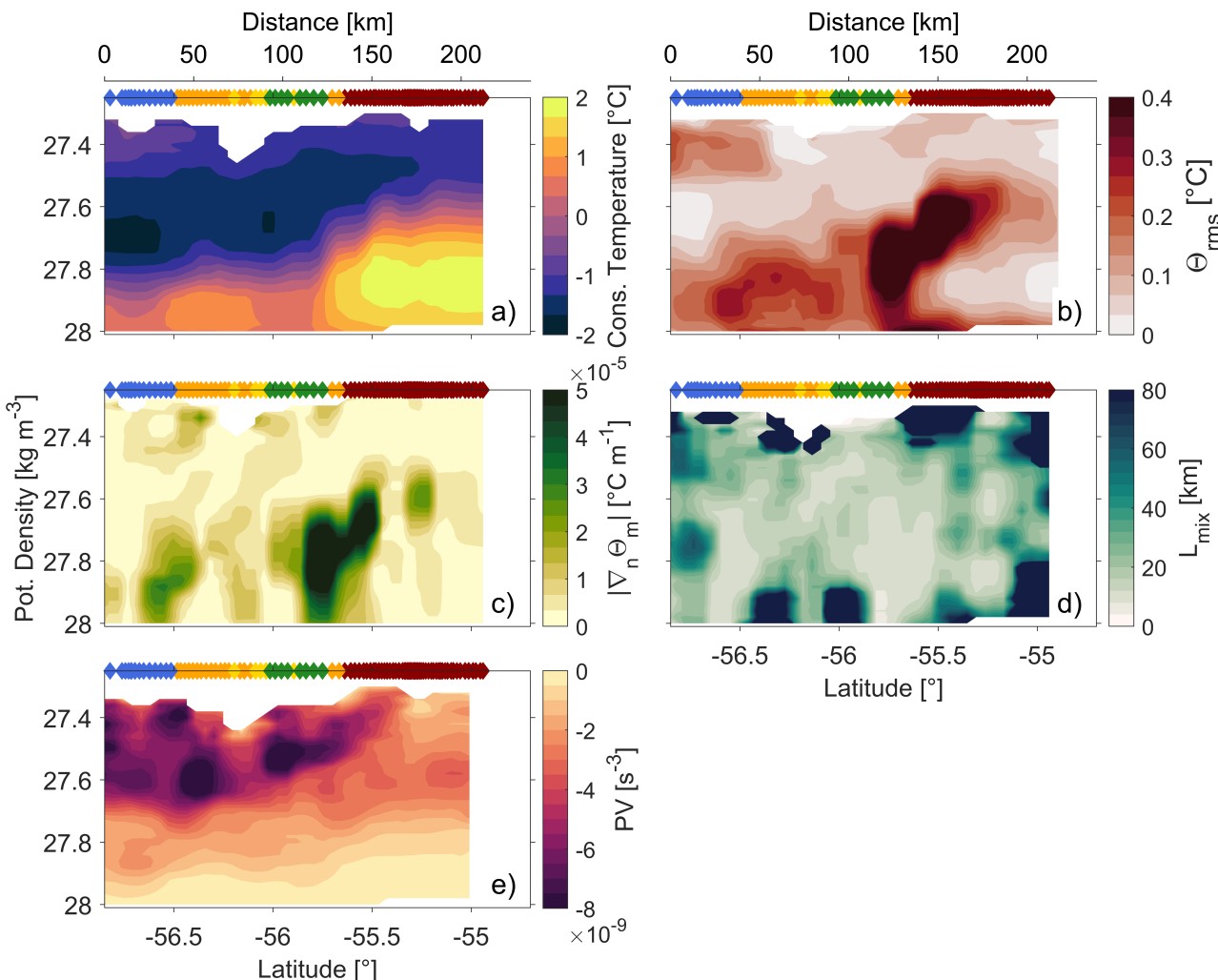

**Figure B1.** As for Fig. 9 but for transect B.

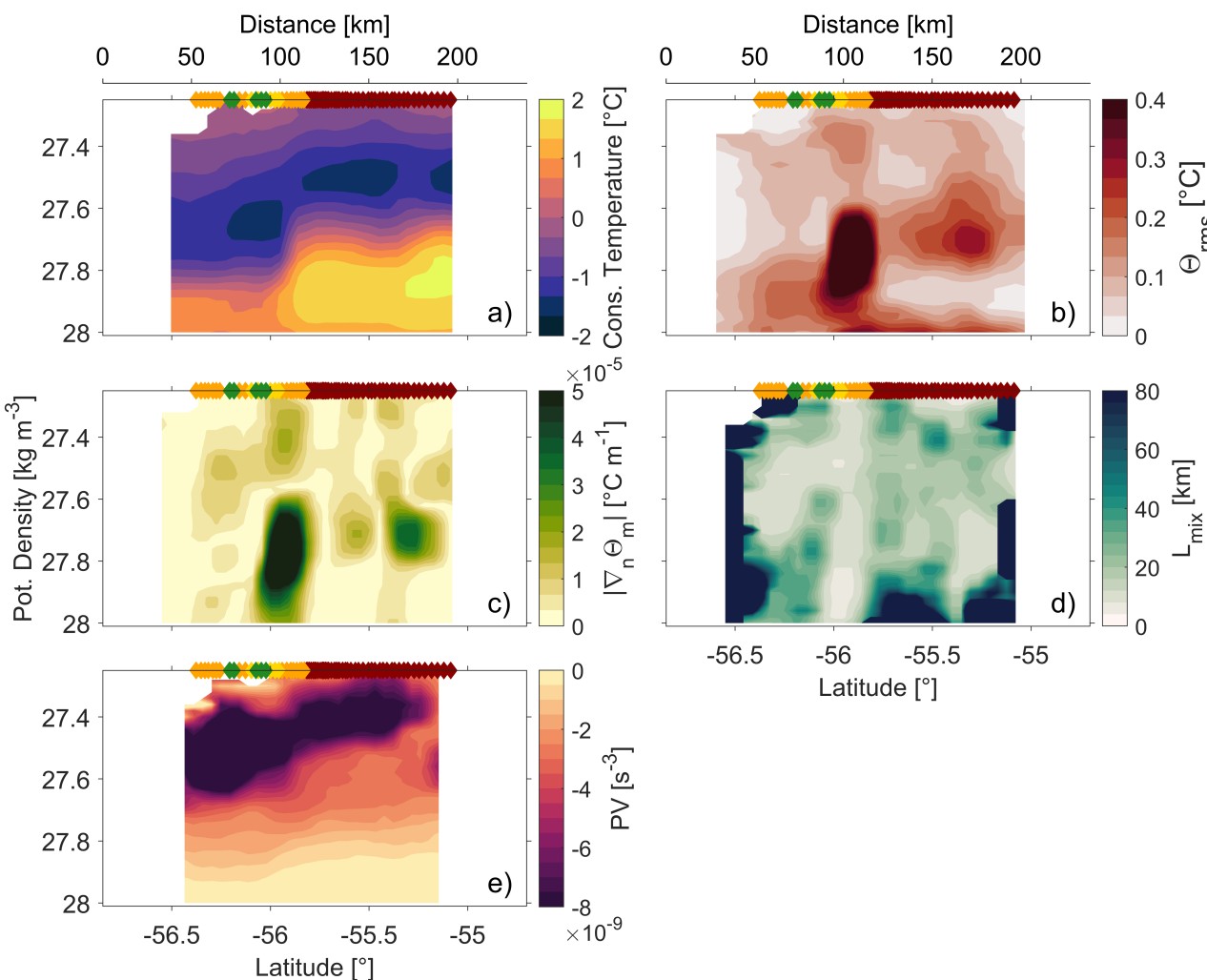

**Figure B2.** As for Fig. 9 but for transect D.

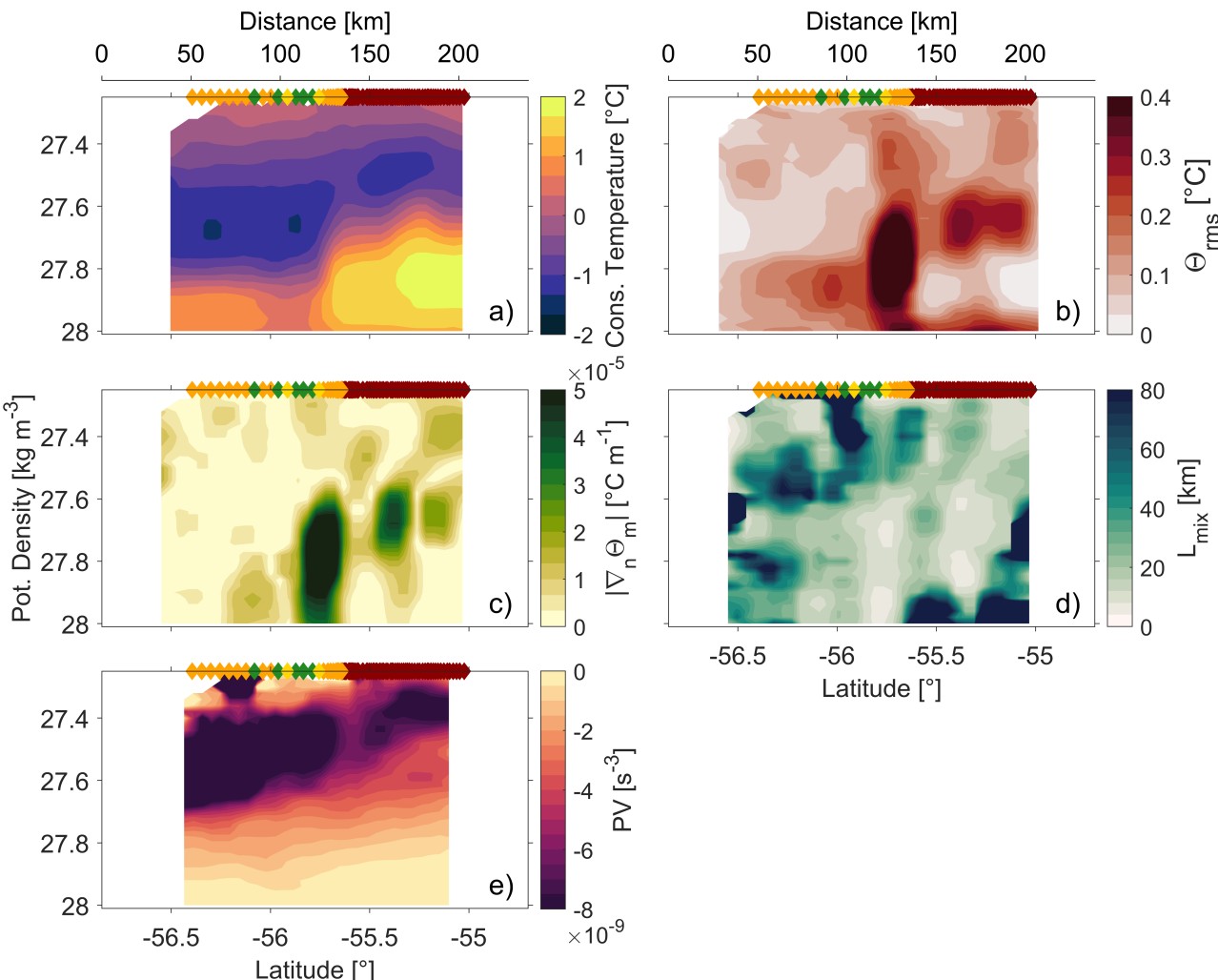

**Figure B3.** As for Fig. 9 but for transect E.

*Author contributions.* RO, KJH and SS developed the concept for the manuscript. RO led the writing process; analysed and processed the glider data. The field experiment design was led by SS and KJH. The gliders were piloted by all authors. KJH, GMD, MdP, SS and LCB provided valuable input and guidance for the development of this study. All authors contributed to the text.

*Competing interests.* KJH is a member of the editorial board of Ocean Science

*Acknowledgements.* The deployment of the glider SG537 and the time of RO, KJH and GMD were supported by funding from the European
Research Council (ERC) under the European Union's Horizon 2020 research and innovation programme (COMPASS, Advanced Grant agreement No. 741120). The deployment of the glider SG640 was supported by the following grants of SS: Wallenberg Academy Fellowship (WAF 2015.0186) and Swedish Research Council (VR 2019-04400). SS and MdP were supported by the European Union's Horizon 2020 research and innovation program under Grant agreement no. 821001 (SO-CHIC). We thank Sea Technology Services (STS), SANAP, the captain and crew of the S.A. Agulhas II for their field-work/ technical assistance associated with the deploying of the gliders in this study.

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
