# Peer review of "Stirring across the Antarctic Circumpolar Current's Southern Boundary at the Greenwich Meridian, Weddell Sea"

_EGUsphere, 2022_

## Referee Comment (RC1)

This study used three months of high-resolution data from glider transects over the Antarctic Circumpolar Current's Southern Boundary to assess its variability in location and intensity in terms of lateral gradients and velocities. The observation indicates that a mesoscale cold-core eddy influences the Southern Boundary's frontal structure by disrupting the temperature transition zone at the subpolar limb, enforcing stronger density gradients across the front and affecting the frontal jet strength. The authors also showed that small mixing length scale and more pronounced PV gradients at the Southern Boundary were concurrent with the cold-core eddy, and the variability of its barrier/blender nature over a multidecadal timescale was discussed.

The presented observation is very attractive and seemingly provides novel findings about the controlling factors of the frontal structure and isopycnal fluxes in the vicinity of the Southern Boundary, the oceanic gateway to the Antarctic coast. The manuscript is well organized, the logic is clear, and the presentation meets necessary and sufficient. Therefore, I strongly support its publication in the journal.

Before publication, however, I have several recommendations and questions about the manuscript as follows:

<major point 1>

I first want to assure what is the frontal jet focused on this study is. Based on the Orsi's temperature criteria, the authors defined the location of SB, and subsequently the SB was re-defined based on the neighbouring ADT contour and its maximum ADT gradient. However, according to Sokolov and Rintoul (2009a) also cited in the manuscript, the corresponding frontal jet seems to be the Southern ACC Front at 56–57S (see the figure below).

[Figure]

**Figure 2.** A typical SSH gradient field south of Africa (11 October 2000) overlaid with the synoptic position of the SSH contours associated with each front (the values of the SSH streamlines corresponding to the fronts are derived for the whole period of altimetry observations). The Southern Ocean fronts are color coded from south to north as follows: SB, black; SACCF (-S and -N), blue; PF (-S, -M, and -N), magenta; SAF (-S, -M, and -N), black; SAZ/STZ (-S, -M, and -N), blue. The middle branches of the fronts are shown by solid lines, while the northern and southern branches (where applicable) are show by dotted lines. In the case of the SACCF, the northern branch is shown by solid line and the southern branch is shown by dotted line. The 2000 m bathymetric contour is shown by thin black line.

Then, how can we call the frontal jet of interest? My recommendation is "to use the SACCF instead of SB". Originally, Orsi+1995 defined the SB as 1.5 degC at T-max based on a fact that the isotherm is well aligned with the poleward limit of oxygen-depleted layer, which is characteristic to UCDW in their dataset. Since UCDW conceptually configures the upper branch of the Southern Ocean MOC, it is natural to define UCDW as the oxygen-depleted layer. In other words, without showing the correspondence between the poleward limit of oxygen-depleted layer and the isotherm, it would be non-trivial to define the position of SB using temperature. Strictly speaking, isopycnal poleward migration of UCDW over decades can change the position of the T-max isotherm independent of the frontal shift and the positional relationship between isotherms and dynamical fronts (e.g., Yamazaki et al., 2021), so that the SB's definition introduced by Orsi+1995 based on the pre-1990's data may not be valid at present. Moreover, as mentioned by the authors, the SB is a water mass boundary and not necessarily accompanied with a frontal jet, whereas the SACCF is a dynamical front by its definition.

<major point 2>

I noticed the mixing length calculation shown in Figs 9 and 10 is substantially different from the convention (e.g., as performed in Naveira Garabato, 2011). In this study, the mean tracer gradient ($\nabla\Theta_m$) seems to be calculated from one temperature section smoothed with twice the baroclinic deformation radius horizontally and 0.08 kg/m-3 vertically, whereas it has conventionally been calculated from the averaged tracer field for repeated observations. As for the hydrographic variability ($\Theta_{rms}$), although I could not fully understand the method, it seems like the difference between the original high-resolution section and the smoothed section in this study, whereas it is conventionally the standard deviation of tracer over the repeated observations (see schematic below; left: convention, right: this study). In this way, the difference in mixing length among the two sections can be discussed as in Figs 9 and 10.

[Figure]

This mixing length calculation and the "hydrography-based" mixing length change are new to me, so it would be very helpful if the authors can provide any reference that adopted the same/similar method. Otherwise, I think more explanation for its validity needs to be provided; for example, how many data points are required to quantify the mixing length over the horizontal scale of interest? Comparison to the mixing length calculated from the conventional scheme (in this study, $\nabla\Theta_m$ is calculated simply from the average of five transects, and $\Theta_{rms}$ is simply the standard deviation over the five transects) and their physical differences? Sensitivity to the choice of the horizontal/vertical smoothing scale?

Please note, the estimate in this study should be more informative than the conventional estimate in a sense that the estimate is expected to be purely affected by the mesoscale features.

<minor points>

L35: I assume the authors want to declare the definition of southern boundary in this study?

L93: "Internal" Rossby radius or "baroclinic deformation radius"? I recommend adding a reference (e.g., Chelton+ 1998, JPO) here as it is also critical to the mixing length calculation.

L110: LCDW should travel poleward beyond the southern boundary as it constitutes the lower MOC to merge with AABW.

L111: "28km" – add "spanning over"?

L118: Fig. 4 – I wonder that the surface drift (cyan) generally seems weaker than the DAC (magenta) despite of the eastward geostrophic shear above 1000m (Figs. 2 and 3). Can you explain why, and which estimate is more reliable?

L131: "south" – replace with "north"? Perhaps providing the horizontal scale of the bowl structure would help understanding.

L133: What is "the coincident changes"?

L143: "40 km" – the baroclinic deformation radius is 10-15km, then we can expect eddy's diameter of 20-30 km?

L145: I could see westward velocities characteristic to the eddy's southern edge by the surface drift and the altimetric velocities, while they are unlikely visible in the DAC.

L148: "advected" – it might also be possible that the eddy was merged with a larger structure (probably, jet's meander) to its west or east.

L150: Then, how sea-level depressions (white contours) larger than the cold eddy can be interpreted?

L161: Absolute salinity needs unit g/kg.

L169: Why the DAC is more appropriate as the reference than the surface drift?

L170: 80 cm/s – this far exceeds the altimetric speed and the surface drift.

L174: "the gradient of ADT (Fig. 8a,c)" – unit is m/m in Fig 8

L177: It also seems like the major front (SACCF-N) and the minor front (SACCF-S) regulate the barrier strength. Can you please provide any effects by jet's meandering?

L184: "strengthens" – does this refers to inverse cascade dynamically?

L204: How the temperature fluctuation is calculated? (This would be why I could not fully understand the calculation)

L203: Strictly speaking, the cross-section (defined by glider positions), along-stream (defined by the streamline), and zonal components are all different. Please elaborate on it throughout the manuscript or demonstrate these differences do not change the result.

L219: "The PV is further considered along potential density surfaces with…" – Simply, "PV is calculated over"? Or, is this meant to be "potential density surfaces are considered to be isoneutral"?

L263: There is section 4.1 but following sections 4.2 etc. are absent.

L275: "The Southern Boundary's location (determined from the frontal jet)" – I recommend to replace with the SACCF.

L287: "In summary" – meridional eddy heat flux may be given by $-k\nabla\Theta$, where k is isopycnal diffusivity associated with the mixing length. Then, how changes in $\nabla\Theta$ affect the meridional heat transport? Is it safely negligible even on account of the offshore warming?

---

## Author Comment (AC1)

Responses to Reviewers

Stirring across the Antarctic Circumpolar Current's Southern Boundary at the Greenwich Meridian, Weddell Sea

We thank the reviewer for their helpful comments and suggestions that have strengthened our paper.  In our responses below, the reviewers' comments are in black, our responses are in blue and the revised text is in purple.

Reviewer 1 – Kaihe Yamazaki
This study used three months of high-resolution data from glider transects over the Antarctic Circumpolar Current's Southern Boundary to assess its variability in location and intensity in terms of lateral gradients and velocities. The observation indicates that a mesoscale cold-core eddy influences the Southern Boundary's frontal structure by disrupting the temperature transition zone at the subpolar limb, enforcing stronger density gradients across the front and affecting the frontal jet strength. The authors also showed that small mixing length scale and more pronounced PV gradients at the Southern Boundary were concurrent with the cold-core eddy, and the variability of its barrier/blender nature over a multidecadal timescale was discussed.

The presented observation is very attractive and seemingly provides novel findings about the controlling factors of the frontal structure and isopycnal fluxes in the vicinity of the Southern Boundary, the oceanic gateway to the Antarctic coast. The manuscript is well organized, the logic is clear, and the presentation meets necessary and sufficient. Therefore, I strongly support its publication in the journal.

Before publication, however, I have several recommendations and questions about the manuscript as follows:

<major point 1>
I first want to assure what is the frontal jet focused on this study is. Based on the Orsi's temperature criteria, the authors defined the location of SB, and subsequently the SB was redefined based on the neighbouring ADT contour and its maximum ADT gradient. However, according to Sokolov and Rintoul (2009a) also cited in the manuscript, the corresponding frontal jet seems to be the Southern ACC Front at 56–57S (see the figure below). Then, how can we call the frontal jet of interest? My recommendation is "to use the SACCF instead of SB". Originally, Orsi+1995 defined the SB as 1.5 degC at T-max based on a fact that the isotherm is well aligned with the poleward limit of oxygen-depleted layer, which is characteristic to UCDW in their dataset. Since UCDW conceptually configures the upper branch of the Southern Ocean MOC, it is natural to define UCDW as the oxygen-depleted layer. In other words, without showing the correspondence between the poleward limit of oxygen-depleted layer and the isotherm, it would be non-trivial to define the position of SB using temperature. Strictly speaking, isopycnal poleward migration of UCDW over decades can change the position of the T-max isotherm independent of the frontal shift and the positional relationship between isotherms and dynamical fronts (e.g., Yamazaki et al., 2021), so that the SB's definition introduced by Orsi+1995 based on the pre-1990's data may not be valid at present. Moreover, as mentioned by the authors, the SB is a water mass boundary and not necessarily accompanied with a frontal jet, whereas the SACCF is a dynamical front by its definition.

We respectfully disagree with the reviewer suggesting that the frontal jet is associated with the Southern ACC Front rather than the Southern Boundary. Previous studies (Billany et al. 2010; Swart et al. 2010) focusing on the fronts of the ACC at the Greenwich Meridian identified the Southern Boundary around 55.5 °S (as in our study), whereas the Southern ACC Front was identified around 53°S. Please see Fig. 1,8 and Table 2 from Swart et al. 2010 for further clarification.

[Figure]

**Figure 1.** Locations of the eight CTD sections used in this study. The AJAX section (blue circles), A21 section (green diamonds), 1992 A12 section (red squares), 1999 A12 section (magenta triangles), 2000 A12 sections (white stars), and 2002 A12 section (white triangles). The solid black line represents the repeat cruise track of the GH CTD and XBT sections. Traces of the ACC fronts, by *Orsi et al.* [1995], and the bathymetry (in m) has been overlaid. STF, Subtropical Front; SAF, Subantarctic Front; APF, Antarctic Polar Front; SACCF, southern ACC front; SBdy, southern boundary of the ACC. The gridded boxes represent the latitudinal zones from which Argo float data were extracted to derive a seasonal model for the region.

[Figure]

**Figure 8.** The mean MADT gradient (in dyn m 100 km$^{-1}$), at the GH line, marks the positions of the ACC fronts (marked and labeled).

**Table 2.** Mean Value of MADT, Used to Follow the Fronts in the MADT Time Series, as Well as the Mean Latitudinal Position of Each Front and Their Standard Deviations Are Listed

| Front | Mean MADT (dyn m) | Front Position (°S) | Standard Deviation (° latitude) |
|---|---|---|---|
| STF | 1.41 | 39.9 | 1.51 |
| SAF | 1.15 | 44.3 | 0.36 |
| APF | 0.49 | 50.4 | 0.27 |
| SACCF | 0.18 | 53.4 | 0.21 |
| SBdy | −0.07 | 55.5 | 0.32 |

Furthermore, Billany et al. (2010) reproduced the ACC front locations from Orsi et al. (1995) and identified the Southern Boundary at a location (around 55.5°S) that agrees with Swart et al (2010) and our study. See Table 1 from Billany et al. (2010) for further justification.

**Table 1**
Criteria used to locate the ACC Fronts, reproduced from Orsi et al. (1995).

| Front | Criteria | Position (°S) defined by Orsi et al. (1995) | MADT-derived mean frontal position (°S) | Frontal position standard deviation (°) | MADT values followed (dyn m) |
|---|---|---|---|---|---|
| STF | 10 °C<θ$_{100m}$<12 °C | 38.4 | 38.5 | 0.56 | 1.56 |
| SAF | S<34.20 at Z<300 m θ>4–5 °C at 400 m | 45.7 | 45.3 | 0.31 | 1.90 |
| APF | θ<2 °C along θ$_{min}$ at Z<200 m | 49.4 | 50.0 | 0.24 | 0.58 |
| SACCF | θ>0 °C along θ$_{min}$ at Z<150 m | 52.4 | 53.5 | 0.19 | 0.19 |
| SBdy | Southern limit of vertical maximum of θ>1.5 °C, (~200 m) | 56.1 | 55.6 | 0.28 | −0.06 |

The fronts in the table are as follows; Subtropical Front (STF), Sub-Antarctic Front (SAF), Antarctic Polar Front (APF), Southern ACC Front (SACCF), Southern Boundary of the ACC (SBdy). θ is the potential temperature, S is the salinity. The positions determined by Orsi et al. (1995) are for the Greenwich Meridian. The MADT-derived mean frontal position and associated standard deviations are given for each front.

The reviewer is correct that the Southern Boundary was originally defined as a water mass boundary. However, a more recent update of this definition clearly showed that the Southern Boundary is associated with the frontal jet at the Greenwich Meridian. Swart et al. (2010) projected hydrographic sections crossing the ACC onto baroclinic stream function space, which provides a two-dimensional gravest empirical mode (GEM). The GEM explained about 97% of the temperature and density variance within the ACC domain. GEM-produced velocities (Fig. 16 of Swar et al. (2010)) compared closely with observations and showed that the Southern Boundary is associated with a frontal jet at around 55.5°S.

[Figure]

**Figure 16.** (a) The sum of the time-averaged (1992–2008) latitudinal distribution of the cross-sectional velocities (in m s$^{-1}$) at the GH line. (b) The vertical distribution of the cross-sectional velocities (in m s$^{-1}$) are depicted at the GH line. The large arrows show the mean positions of the major ACC fronts, identified by the ADT, while smaller arrows show those additional jet-like structures that are not clearly seen in the MADT velocities, presented in Figure 9.

The reviewer further argued that a characteristic of UCDW is the oxygen-depleted layer. Our glider data provide oxygen data (see examples for Transect A and C below) and show that oxygen is depleted within the UCDW layer.

[Figure]

All above stated findings provide justification that the transition in water mass properties and frontal jet that we discuss within this study, is associated with the Southern Boundary of the ACC.

Additional discussion of the above mentioned citations and definitions of the Southern Boundary have been added to the manuscript for further clarification.

<major point 2>
I noticed the mixing length calculation shown in Figs 9 and 10 is substantially different from the convention (e.g., as performed in Naveira Garabato, 2011). In this study, the mean tracer gradient ($\nabla\Theta\_m$) seems to be calculated from one temperature section smoothed with twice the baroclinic deformation radius horizontally and 0.08 kg/m-3 vertically, whereas it has conventionally been calculated from the averaged tracer field for repeated observations. As for the hydrographic variability ($\Theta\_{rms}$), although I could not fully understand the method, it seems like the difference between the original high-resolution section and the smoothed section in this study, whereas it is conventionally the standard deviation of tracer over the repeated observations (see schematic below; left: convention, right: this study). In this way, the difference in mixing length among the two sections can be discussed as in Figs 9 and 10.

This mixing length calculation and the "hydrography-based" mixing length change are new to me, so it would be very helpful if the authors can provide any reference that adopted the same/similar method. Otherwise, I think more explanation for its validity needs to be provided; for example, how many data points are required to quantify the mixing length over the horizontal scale of interest? Comparison to the mixing length calculated from the conventional scheme (in this study, $\nabla\Theta\_m$ is calculated simply from the average of five transects, and $\Theta\_{rms}$ is simply the standard deviation over the five transects) and their physical differences? Sensitivity to the choice of the horizontal/vertical smoothing scale?

Please note, the estimate in this study should be more informative than the conventional estimate in a sense that the estimate is expected to be purely affected by the mesoscale features ???

The reviewer is correct that our method differs slightly from the method used by Naveira Garabato et al. (2011). It has to be mentioned here that their study used ship-based hydrographic sections rather than our closely-spaced glider sections. Our study is based on a method for glider data by Dove et al. (2023) & Viglione (PhD Thesis). We have added this reference to our study to further justify the method that we used for glider data. The aim here is to provide a 'large scale' temperature field by smoothing over twice the Rossby Radius. Thus, the $\Theta\_m$ is not based on an average between the transects but rather a smoothed 'large scale' version of the high-resolution temperature data, where $\Theta\_{rms}$ is the standard deviation between the 'large scale' and the high resolution temperature field. Furthermore, the mixing length scale contrast between Transect A and C are sufficiently larger (about 6 times) than the scale of the observations (5 km), indicating the capability of our highly-resolved sections to reveal the mixing length scale contracts between sections.

L35: I assume the authors want to declare the definition of southern boundary in this study?
Yes, the definition of the Southern Boundary is defined in L35 via water mass properties. We now added the additional discussion of the Southern Boundary associated with a frontal jet after Swart et al. (2010) as well as its location at the Greenwich Meridian (around 55.5°S). Please see response to major point 1 for further details.

L93: "Internal" Rossby radius or "baroclinic deformation radius"? I recommend adding a reference (e.g., Chelton+ 1998, JPO) here as it is also critical to the mixing length calculation.

We refer to the Rossby Radius of deformation (baroclinic deformation radius). Suggested reference has been added.

L110: LCDW should travel poleward beyond the southern boundary as it constitutes the lower MOC to merge with AABW.

LCDW is not detected south of the Southern Boundary in the observations in this study. The glider transects show that LCDW underneath UCDW north of the Southern Boundary, but not beyond the Southern Boundary. Therefore, mentioning the southward extent of LCDW across the Southern Boundary here would be speculation and has thus not been added.

L111: "28km" – add "spanning over"?
The suggestion has been added to L111.

L118: Fig. 4 – I wonder that the surface drift (cyan) generally seems weaker than the DAC (magenta) despite of the eastward geostrophic shear above 1000m (Figs. 2 and 3). Can you explain why, and which estimate is more reliable?

We thank the reviewer for spotting this error. There has been a typo in the Fig. caption. The cyan colors show the DAC and the magenta colors show the surface drift. This has been corrected in the manuscript.
The geostrophic velocities (Fig. 8) are surface intensified, which suggests that the surface drift should be larger than the DAC (which it is). The winds above the Southern Boundary usually have a west to east orientation, so would tend to further increase the surface drift, which additionally explains the difference between surface drift and DAC.
Therefore the surface drift is influenced by surface currents and winds, whereas the DAC contains information of the deeper water column (1000 m).

L131: "south" – replace with "north"? Perhaps providing the horizontal scale of the bowl structure would help understanding.
The 'south' in L131 refers to the location of the bowl structure, rather than the occurrence of warmer waters north of the Southern Boundary. Depending on the defined location of the Southern Boundary (southernmost limit of UCDW) the bowl structure would still be south of the Southern Boundary. This sentence has been edited to improve readability. Horizontal scale description (latitude) of the bowl structure has been added as well.

L133: What is "the coincident changes"?
The 'coincident changes' here refer to the characteristics, such as water mass properties and bowl structure south of the Southern Boundary that match for the transects (A, B, D and E) which do not necessarily match for transect C. We have adapted the sentence to clarify.

L143: "40 km" – the baroclinic deformation radius is 10-15km, then we can expect eddy's diameter of 20-30 km?
Yes, apologies for that. The eddies are about 20-30 km in diameter.

L145: I could see westward velocities characteristic to the eddy's southern edge by the surface drift and the altimetric velocities, while they are unlikely visible in the DAC.
We apologize for that. The westward velocities at the eddy's southern edge are visible in the DAC too. We have adjusted the arrow size in Fig. 4 to increase visibility.

L148: "advected" – it might also be possible that the eddy was merged with a larger structure (probably, jet's meander) to its west or east.
Yes, absolutely. We have added your suggestion to L148.

L150: Then, how sea-level depressions (white contours) larger than the cold eddy can be interpreted?
We are not sure what the reviewer is referring to here? We are assuming that the reviewer is referring to Fig. 4. The other white contours here refer to other cold-core eddies interacting with the Southern Boundary/ SACCF. In Fig. 4 the white contours depict the transition zone from eddy core towards the outside of the eddy. We have added that explanation to the discussion in the text to clarify that.

L161: Absolute salinity needs unit g/kg.
Units have been added to absolute salinity.

L169: Why the DAC is more appropriate as the reference than the surface drift?
Please see the response to L118.

L170: 80 cm/s – this far exceeds the altimetric speed and the surface drift.
Yes, this is quite a common issue. With regards to the surface drift please see L118. Satellite altimetry- derived currents are necessarily temporally and spatially smoothed by the process of creating the gridded product from relatively widely-spaced altimetric tracks infrequently repeated. This may lead to eddies and front being in the correct location, but averaged/smoothed in e.g. current speed so that values from satellite altimetry tend to be smaller than observed current speeds. We have added the following lines to the text to emphasize that in more detail.

L174: "the gradient of ADT (Fig. 8a,c)" – unit is m/m in Fig 8
Yes, thanks for spotting that. Unit has been corrected.

L177: It also seems like the major front (SACCF-N) and the minor front (SACCF-S) regulate the barrier strength. Can you please provide any effects by jet's meandering?
Between transect A and C the frontal jet of the Southern Boundary has shifted meridionally. The location of the Southern Boundary has been discussed in L104-L124. Additional discussion with respect to possible influences on the barrier strength due to the jets meandering has been added.

L184: "strengthens" – does this refers to inverse cascade dynamically?
We are just referring to the changing density gradients here. We have changed L184 to 'amplifies' to avoid confusion.

L204: How the temperature fluctuation is calculated? (This would be why I could not fully understand the calculation)
The temperature root mean square $\Theta_{rms}$ is calculated as the standard deviation of the temperature anomalies from the mean 'large scale' temperature field and the high resolution temperature field ($\Theta_m - \Theta$). This has been added to the method description.

L203: Strictly speaking, the cross-section (defined by glider positions), along-stream (defined by the streamline), and zonal components are all different. Please elaborate on it throughout the manuscript or demonstrate these differences do not change the result.
First the glider locations are projected onto a meridional line and then we calculate the geostrophic shear. Thus, we only calculate the zonal velocity component. Furthermore, we find in the key transects (A and C) the flow at the Southern Boundary and over the associated frontal jet is zonal.

L219: "The PV is further considered along potential density surfaces with…" – Simply, "PV
is calculated over"? Or, is this meant to be "potential density surfaces are considered to be
isoneutral"?
According to the reviewer's suggestion the sentence has been changed to:
The PV is calculated over potential density surfaces with …

L263: There is section 4.1 but following sections 4.2 etc. are absent.
We thank the reviewer for spotting this error. Section 4 is now divided into section 4.1 and section 4.2.

L275: "The Southern Boundary's location (determined from the frontal jet)" – I recommend
to replace with the SACCF.
Please see response to major comment 1.

L287: "In summary" – meridional eddy heat flux may be given by -k$\nabla\Theta$, where k is isopycnal diffusivity associated with the mixing length. Then, how changes in $\nabla\Theta$ affect the
meridional heat transport? Is it safely negligible even on account of the offshore warming?
This is a really good question that we will give careful thought, when we prepare the revised manuscript.
 (The argument is here based on the gradients across the front, which include temperature gradients. Studies (Shi et al (2021)) have shown that due to upper ocean warming the gradients across the main ACC fronts are amplified. In cases

where barrier properties are enhanced (due to amplified gradients) the diffusivities across are near 0, thus the eddy heat transport at least has the potential to become very small).

---

## Author Comment (AC2)

Responses to Reviewers

Stirring across the Antarctic Circumpolar Current's Southern Boundary at the Greenwich Meridian, Weddell Sea

We thank the reviewer for their helpful comments and suggestions that have strengthened our paper.  In our responses below, the reviewers' comments are in black, our responses are in blue and the revised text is in purple.

Reviewer 2

5 repeated glider surveys across the Southern Boundary (SBDY) of Antarctic Circumpolar Current are used to investigate SBDY's cross-frontal behaviours under eddy and non-eddy regimes. Eddy presence enhances cross-frontal density gradient supressing the cross-frontal mixing whereas eddy absence, comparing to eddy presence, is accompanied by a weaker cross-frontal density gradient. These results are interpreted under the context of a multidecadal evolution of SBDY speed/location derived from satellite data. Authors concluded that the enhanced eddy activities and accelerated SBDY are occurring at the same time in opposition in affecting the meridional exchanges of tracers cross SBDY at Greenwich Meridian. I found this work is interesting and potentially important for the community in understanding the Weddell Gyre heat content evolution under the context of climate changes.

I have one concern about this manuscript. This work highlights that the different cross-fontal properties are associated with eddy presences exemplified by comparing transect A and transect C. These contrasting results between eddy and non-eddy regimes need to be strengthened by a quantified uncertainty that could be raised from different glider sampling intensity along the transect because it seems to me that the transect C does not take profiles as frequently as transect A by looking at the station distribution from two transects. See also the relevant comments below. I am happy to see this manuscript published once my concerns herein are addressed properly.

General comments:

1. Most results present in this manuscript based on the comparison between transect A and transect C, where the authors argue that eddy presence/absence is the reason for the observe difference. The glider station (marked as triangle on top of cross-section plot, most evident in Figure 8) distribution between A and C is different. Can author quantify the potential uncertainty caused by different glider station distributions on the present cross-frontal difference?

   Yes, the reviewer is correct that the data are sampled in a higher horizontal resolution in Transect A (290 vertical profiles, 145 dives), whereas Transect C has a lower horizontal resolution (92 vertical profiles, 46 dives). However, within this study the difference in glider station distribution is negligible as all data are horizontally gridded onto a uniform grid. Thus, for all calculations and final results the horizontal and vertical resolution of all transects considered is uniform. We further tested a subsampling of Transect A (bootstrapping

method) with the number of profiles of Transect C and found that key characteristics in Transect A remained unchanged. Specifically for Fig. 8 we have changed the diamonds at the top of each transects to show that the grid for Transect A and C is uniform. We have added the above mentioned information to the caption of Fig. 8 (see below) as well as in the text of the manuscript to clarify for the readers.

[Figure]

Figure 8. Real-time altimetric ADT and gradients of ADT ($\nabla y$ ADT) for (a) transect A and (c) transect C. (b,d) Geostrophic velocities perpendicular to the respective glider transects A and C and referenced to the DAC with a horizontal smoothing (moving mean filter) of approx. 15 km (Rossby radius within the region of interest). Positive geostrophic velocities are defined as eastwards (red). Black contours are as in Fig. 2. The black diamonds at the top of each panel show the uniform horizontal gridding with 5 km spacing of

transect A and C. The dashed black lines indicate the location of the Southern Boundary based on the southernmost strong ADT gradient.

2. This may or may not be resolved by typesetting, but I found that quite a few figures are far from where they were discussed. For example, section 3 in page 8 discussed Figure 4 to Figure 8, while Figure 8 is displayed at Page 14. I suggest authors to condense down figure volume, such as, putting multiple subpanels into one integrated figure, leaving the results for transect B, D, E in Supp Mats as they were barely mentioned, T-S plots with highlighted regimes taking up one subpanel spaces can be replaced by combining mainly discussed regimes in one T-S plot and mask other data points with grey colour, etc.

We suggest merging Fig. 6 and 7 to reduce Fig. volume. The figures can surely be brought closer to where they are discussed in the manuscript. This is a matter of the typesetting of the final article. We will raise your concerns if the figures are still poorly placed when we receive the proof of the article. With respect to Figs. 1 and 2 we think that transects B, D, E should still be included to introduce the entire data set and to justify why we focus on Transects A and C later.

Specific comments:

L6: 'quite rapid'→'high-frequency' or 'transient'?
Quite rapid has been changed to transient.

L35: delete 'globally', the word 'globally' is misplaced as the SBDY is not a global feature, is it? 'Climatologically' is sufficient here.
This has been changed as suggested by the reviewer.

L42: '…further represent the southernmost boundary to mixing'. I found this sentence a bit ambiguous… I believe that the mixing process in general is happening everywhere, and I don't think authors have set the context of using the term mixing to refer the cross-frontal mixing happened at the SBDY.
This sentence has been edited according to the reviewers concern. The sentence now reads as:
The frontal jets of the ACC are often seen as barriers to meridional horizontal mixing (e.g. Naveira Garabato et al. (2011)). The frontal jet associated with the Southern Boundary, as the southernmost of the ACC frontal jets, marks the boundary between the northern limit of sea ice formation and the ACC.

L60:'The majority of studies almost entirely…', need refs here or is author referring to aforementioned studies? If so, please indicate.
Yes, all aforementioned studies are referred to here. This has been edited in the manuscript as suggested by the reviewer.

L160: 'converge'. The T-S plots do not show this 'convergence' particularly clear. Adding arrows to indicate this in the T-S plots.
Edited according to the reviewer's suggestion.

L164-167: I do not fully understand this. The similarity of the properties between eddy and south of SBDY suggest eddy originated from south of SBDY, okay, then what is the meaning of mentioning the slight temperature/salinity difference below/above the thermocline? Plus, why do authors mention the salinity difference in reference to thermocline?

We have adjusted the sentence to clarify:

The clockwise eddy identified in transect A (Fig. 6 a,d,e) presents properties similar to the cold regime but with slightly higher temperatures (about 0.4 to 0.6◦C higher) below the thermocline and slightly reduced salinities above the depth of the thermocline. Note that the eddy is surface intensified and therefore changes in the surface properties are expected, although the eddy is more clearly identified in the sub-thermocline temperatures and salinities. The similar water mass properties of the eddy and the cold regime suggest that the eddy originated south of the Southern Boundary.

L179-182: Mention the criteria and the table somewhere earlier in the section. This section has covered many figures that use such color-coding criteria. Best to mention it in the first place to avoid confusion for readers.

We agree with the reviewer and have moved the table with the criteria earlier in the section to improve clarity.

L185: the eddy passage could be one of the reasons for the difference in horizontal density gradient between transect A and C. Figure 8 shows a smooth ADT for C and rough ADT for A which makes me realize that the profiling intensity of A and C is also different. Transect A has more profiles in general than Transect C across the front. Does this fact play any role? Authors should quantify the uncertainty on horizontal density gradient caused by different sampling intensity by subsampling a high-res model results/reanalysis or any other sensible measures.

Please see response to major point 1.

L204: It is not clearly stated how the temperature fluctuation, θ', is computed.

The temperature root mean square $\Theta_{rms}$ is calculated as the standard deviation of the temperature anomalies from the mean 'large scale' temperature field and the high resolution temperature field ($\Theta_m - \Theta$). This has been added to the method description.

L285: The discussion on the long-term behavior of the SBDY and its core speed is sufficiently supported by literatures. However, the sea ice extent seems to be a bit out of place here. I suggest authors to either specify the reason for examining sea ice extent and discuss it extensively in the context of past literatures or simply not to show the sea ice extent at all since it does not correlate well with the available data here and authors just briefly mentioned it… Sea ice advancing and retreats on yearly basis is also controlled by large-scale wind variability, thermal forcing and also internal sea ice dynamic, so it perhaps requires some extra effort to decipher sea ice extent in the context of enhanced frontal jet.

According to the suggestion of the reviewer we have removed the sea ice extent from Fig. 11 as there is currently not enough literature to support our findings and the lack of correlation between frontal jet speed and sea ice extent does not provide

enough evidence to further discuss the sea ice extent within this manuscript. We have further removed L44-45 and L285 from the manuscript.

L294: If authors are referring to the positive SLA blobs into the 2010s, then perhaps the phrase 'anti-cyclonic eddies' is more appropriate than warm core eddies? Studies have shown that not all anti-cyclonic eddies have a coherent warm core structure throughout the vertical extent.
The expression warm core eddy has been replaced with anti-cyclonic eddy.

L309: '…. consistent with Williams et al. (2007) who demonstrated …'
Edited according to the reviewers suggestion.

L301: '… in all transects …', authors mainly discussed transects A, relevant results for B, D, E should be included at least in Supp Mats to make this claim.
Relevant results for B, D and E have been added to the supplementary material.

---

## Author Response (AR1)

Responses to Reviewers

Stirring across the Antarctic Circumpolar Current's Southern Boundary at the Greenwich Meridian, Weddell Sea

We thank the reviewer for their helpful comments and suggestions that have strengthened our paper.  In our responses below, the reviewers' comments are in black, our responses are in blue and the revised text is in purple.

Reviewer 1 – Kaihe Yamazaki

This study used three months of high-resolution data from glider transects over the Antarctic Circumpolar Current's Southern Boundary to assess its variability in location and intensity in terms of lateral gradients and velocities. The observation indicates that a mesoscale cold-core eddy influences the Southern Boundary's frontal structure by disrupting the temperature transition zone at the subpolar limb, enforcing stronger density gradients across the front and affecting the frontal jet strength. The authors also showed that small mixing length scale and more pronounced PV gradients at the Southern Boundary were concurrent with the cold-core eddy, and the variability of its barrier/blender nature over a multidecadal timescale was discussed.

The presented observation is very attractive and seemingly provides novel findings about the controlling factors of the frontal structure and isopycnal fluxes in the vicinity of the Southern Boundary, the oceanic gateway to the Antarctic coast. The manuscript is well organized, the logic is clear, and the presentation meets necessary and sufficient. Therefore, I strongly support its publication in the journal.

Before publication, however, I have several recommendations and questions about the manuscript as follows:

<major point 1>
I first want to assure what is the frontal jet focused on this study is. Based on the Orsi's temperature criteria, the authors defined the location of SB, and subsequently the SB was redefined based on the neighbouring ADT contour and its maximum ADT gradient. However, according to Sokolov and Rintoul (2009a) also cited in the manuscript, the corresponding frontal jet seems to be the Southern ACC Front at 56–57S (see the figure below). Then, how can we call the frontal jet of interest? My recommendation is "to use the SACCF instead of SB". Originally, Orsi+1995 defined the SB as 1.5 degC at T-max based on a fact that the isotherm is well aligned with the poleward limit of oxygen-depleted layer, which is characteristic to UCDW in their dataset. Since UCDW conceptually configures the upper branch of the Southern Ocean MOC, it is natural to define UCDW as the oxygen-depleted layer. In other words, without showing the correspondence between the poleward limit of oxygen-depleted layer and the isotherm, it would be non-trivial to define the position of SB using temperature. Strictly speaking, isopycnal poleward migration of UCDW over decades can change the position of the T-max isotherm independent of the frontal shift and the positional relationship between isotherms and dynamical fronts (e.g., Yamazaki et al., 2021), so that the SB's definition introduced by Orsi+1995 based on the pre-1990's data may not be valid at present. Moreover, as mentioned by the authors, the SB is a water mass boundary and not necessarily accompanied with a frontal jet, whereas the SACCF is a dynamical front by its definition.

We respectfully disagree with the reviewer suggesting that the frontal jet is associated with the Southern ACC Front rather than the Southern Boundary. The reviewer is correct that the Southern Boundary is traditionally defined as a water mass boundary. In this study, we have defined the location of the Southern Boundary according to Orsi et al. (1995). Our data show water mass properties of UCDW ($\Theta > 1.5°C, SA > 34.5\ g\ kg^{-1}$) associated with an oxygen depleted layer (Fig. 1). We apologize for not including the oxygen data in the paper previously, which may have led to some confusion. As for temperatures and salinities, the dissolved oxygen also shows strongest standard deviations at the location of the Southern Boundary between 55.5-56°S (Fig. 2). This definition and additional information have been added to the manuscript (L116-131 and L139-140) to clarify the specific properties of the Southern Boundary for the readers and provide further evidence that we are indeed investigating the Southern Boundary in this study. Note that the dissolved oxygen data were added to Appendix A.

[Figure]

Fig 1. Dissolved oxygen of glider transects A-E. Potential density contours of 27.73 $kg\,m^{-3}$ and 27.93 $kg\,m^{-3}$ are shown in black.The 1.5°C isotherm is shown in white. The triangles at the bottom of each panel, and the black dashed line extending upwards from each triangle, indicate the location of the Southern Boundary as defined by Orsi et al. (1995). The triangles are coloured for each individual transect as in Fig. 1, and the same transect colour coding is used in Fig. 3 of the manuscript. The colours at the top of each panel represent our classification into areas north of the Southern Boundary (red), within a transition zone (orange), within the core of an eddy (green) and on the outer edges of an eddy (yellow), and south of the Southern Boundary (blue). This colour coding is discussed in section 3 of the manuscript.

[Figure]

Fig. 2 Mean (a,c) and standard deviation (b,d) of all glider transects A-E for (a,b) dissolved oxygen. The coloured triangles at the bottom of panel (b), and black dashed lines extending upwards from them are as in Fig. 2 of the manuscript. Data from each transect are binned to the same 5 km horizontal grid and then averaged (mean) for all transects. Partially shaded areas on (a) indicate areas that do not have data from all transects. Mean isopycnals 27.73 $kg\,m^{-3}$ and 27.93 $kg\,m^{-3}$ are shown in black. The mean 1.5 °C isotherm is shown in white.

Although the Southern Boundary is defined purely as a water mass boundary, it has been shown that the Southern Boundary is associated with a frontal jet at the Greenwich Meridian and at other longitudes. Swart et al. (2010) projected hydrographic sections crossing the ACC onto baroclinic stream function space, which provides a two-dimensional gravest empirical mode (GEM). The GEM explained about 97% of the temperature and density variance within the ACC domain. GEM-produced velocities (Fig. 16 of Swart et al. (2010)) compared closely with observations and showed that the Southern Boundary is associated with a frontal jet at around 55.5°S.

[Figure]

**Figure 16.** (a) The sum of the time-averaged (1992–2008) latitudinal distribution of the cross-sectional velocities (in m s$^{-1}$) at the GH line. (b) The vertical distribution of the cross-sectional velocities (in m s$^{-1}$) are depicted at the GH line. The large arrows show the mean positions of the major ACC fronts, identified by the ADT, while smaller arrows show those additional jet-like structures that are not clearly seen in the MADT velocities, presented in Figure 9.

Additionally, previous studies (Billany et al. 2010; Swart et al. 2010) focusing on the fronts of the ACC at the Greenwich Meridian identified the Southern Boundary around 55.5 °S (as in our study), whereas the Southern ACC Front was identified around 53°S. Please see Fig. 1,8 and Table 2 from Swart et al. 2010 for further clarification.

[Figure]

**Figure 1.** Locations of the eight CTD sections used in this study. The AJAX section (blue circles), A21 section (green diamonds), 1992 A12 section (red squares), 1999 A12 section (magenta triangles), 2000 A12 sections (white stars), and 2002 A12 section (white triangles). The solid black line represents the repeat cruise track of the GH CTD and XBT sections. Traces of the ACC fronts, by *Orsi et al.* [1995], and the bathymetry (in m) has been overlaid. STF, Subtropical Front; SAF, Subantarctic Front; APF, Antarctic Polar Front; SACCF, southern ACC front; SBdy, southern boundary of the ACC. The gridded boxes represent the latitudinal zones from which Argo float data were extracted to derive a seasonal model for the region.

[Figure]

**Figure 8.** The mean MADT gradient (in dyn m $100 \text{ km}^{-1}$), at the GH line, marks the positions of the ACC fronts (marked and labeled).

**Table 2.** Mean Value of MADT, Used to Follow the Fronts in the MADT Time Series, as Well as the Mean Latitudinal Position of Each Front and Their Standard Deviations Are Listed

| Front | Mean MADT (dyn m) | Front Position (°S) | Standard Deviation (° latitude) |
|---|---|---|---|
| STF | 1.41 | 39.9 | 1.51 |
| SAF | 1.15 | 44.3 | 0.36 |
| APF | 0.49 | 50.4 | 0.27 |
| SACCF | 0.18 | 53.4 | 0.21 |
| SBdy | −0.07 | 55.5 | 0.32 |

Furthermore, Billany et al. (2010) reproduced the ACC front locations from Orsi et al. (1995) and identified the Southern Boundary at a location (around 55.5°S) that agrees with Swart et al. (2010) and our study. See Table 1 from Billany et al. (2010) for further justification.

**Table 1**
Criteria used to locate the ACC Fronts, reproduced from Orsi et al. (1995).

| Front | Criteria | Position (°S) defined by Orsi et al. (1995) | MADT-derived mean frontal position (°S) | Frontal position standard deviation (°) | MADT values followed (dyn m) |
|---|---|---|---|---|---|
| STF | 10 °C<$\theta_{100m}$<12 °C | 38.4 | 38.5 | 0.56 | 1.56 |
| SAF | S<34.20 at Z<300 m | 45.7 | 45.3 | 0.31 | 1.90 |
| | $\theta$>4–5 °C at 400 m | | | | |
| APF | $\theta$<2 °C along $\theta_{min}$ at Z<200 m | 49.4 | 50.0 | 0.24 | 0.58 |
| SACCF | $\theta$>0 °C along $\theta_{min}$ at Z<150 m | 52.4 | 53.5 | 0.19 | 0.19 |
| SBdy | Southern limit of vertical maximum of $\theta$>1.5 °C, (~200 m) | 56.1 | 55.6 | 0.28 | −0.06 |

The fronts in the table are as follows; Subtropical Front (STF), Sub-Antarctic Front (SAF), Antarctic Polar Front (APF), Southern ACC Front (SACCF), Southern Boundary of the ACC (SBdy). $\theta$ is the potential temperature, S is the salinity. The positions determined by Orsi et al. (1995) are for the Greenwich Meridian. The MADT-derived mean frontal position and associated standard deviations are given for each front.

In this study, we also used ADT gradients from satellite altimetry to locate the frontal jet, which as per definition does not have to be in exactly the same location as the Southern Boundary. The results show that the frontal jet's location, estimated from ADT gradients, is approximately 8 to 30 km to the south of the water mass-based definition of the Southern Boundary. Additional discussion of the above mentioned citations and definitions of the Southern Boundary have been added to the manuscript for further clarification (L140-154).

<major point 2>
I noticed the mixing length calculation shown in Figs 9 and 10 is substantially different from the convention (e.g., as performed in Naveira Garabato, 2011). In this study, the mean tracer gradient ($\nabla\Theta\_m$) seems to be calculated from one temperature section smoothed with twice the baroclinic deformation radius horizontally and 0.08 kg/m-3 vertically, whereas it has conventionally been calculated from the averaged tracer field for repeated observations. As for the hydrographic variability ($\Theta\_rms$), although I could not fully understand the method, it seems like the difference between the original high-resolution section and the smoothed section in this study, whereas it is conventionally the standard deviation of tracer over the repeated observations (see schematic below; left: convention, right: this study). In this way, the difference in mixing length among the two sections can be discussed as in Figs 9 and 10.

This mixing length calculation and the "hydrography-based" mixing length change are new to me, so it would be very helpful if the authors can provide any reference that adopted the same/similar method. Otherwise, I think more explanation for its validity needs to be provided; for example, how many data points are required to quantify the mixing length over the horizontal scale of interest? Comparison to the mixing length calculated from the conventional scheme (in this study, $\nabla\Theta\_m$ is calculated simply from the average of five transects, and $\Theta\_rms$ is simply the standard deviation over the five transects) and their physical differences? Sensitivity to the choice of the horizontal/vertical smoothing scale?

Please note, the estimate in this study should be more informative than the conventional estimate in a sense that the estimate is expected to be purely affected by the mesoscale features ???

The reviewer is correct that our method differs slightly from the method used by Naveira Garabato et al. (2011). It has to be mentioned here that their study used ship-based hydrographic sections rather than our closely-spaced glider sections. Our study is based on a method for glider data described in detail by Dove et al. (2023) & Viglione (PhD Thesis). We have added these references to our study to further justify the method that we used for glider data. The aim here is to provide a 'large scale' temperature field by smoothing over twice the Rossby Radius. Thus, the $\Theta_m$ is not based on an average between the transects but rather a smoothed 'large scale' version of the high-resolution temperature data that we defined as $\Theta_\rho$, where $\Theta\_rms$ is the root mean square difference. We have added the description of the $\Theta_{rms}$ calculation to the manuscript. The lines read as follows:
L240-244: Finally, at each grid point we find the root mean square difference $\Theta_{rms}$ between the value of $\Theta_m$ at that grid point, and the values of $\Theta_\rho$ within a 5-element window in the horizontal (i.e., on the same density surface) centered on that grid point. In other words,

$$\Theta_{rms,i} = \sqrt{\frac{\sum_{j=i-2}^{j=i+2}(\Theta_{\rho,j}-\Theta_{m,i})^2}{5}}$$

where $i$ is an index from south to north along a potential density surface.

L35: I assume the authors want to declare the definition of southern boundary in this study?

Yes, the Southern Boundary is defined in L116-131 with water mass properties as defined by Orsi et al. (1995). We now added the additional discussion of the Southern Boundary associated with a frontal jet after Swart et al. (2010) as well as its location at the Greenwich Meridian (around 55.5°S) to L140-154. Please see response to major point 1 for further details.

L93: "Internal" Rossby radius or "baroclinic deformation radius"? I recommend adding a reference (e.g., Chelton+ 1998, JPO) here as it is also critical to the mixing length calculation.

We refer to the Rossby Radius of deformation (baroclinic deformation radius) and have clarified this in L102-104. Suggested reference has been added.

L110: LCDW should travel poleward beyond the southern boundary as it constitutes the lower MOC to merge with AABW.

The reviewer is correct. We have changed the sentence as follows:
L134-137: Antarctic Surface Water (AASW) occupies the top 150-200 m. To the north of the Southern Boundary, the AASW lies above Upper Circumpolar Deep Water (UCDW, 200-750 m), which in turn lies above Lower Circumpolar Deep Water (LCDW). To the south of the Southern Boundary, LCDW is found higher in the water column, below the AASW.

L111: "28km" – add "spanning over"?

The suggestion has been added to L129.

L118: Fig. 4 – I wonder that the surface drift (cyan) generally seems weaker than the DAC (magenta) despite of the eastward geostrophic shear above 1000m (Figs. 2 and 3). Can you explain why, and which estimate is more reliable?

We thank the reviewer for spotting this error. There was a typo in the Fig. caption. The cyan colors show the DAC and the magenta colors show the surface drift, so the surface drift is indeed larger than the DAC. This has been corrected in the manuscript.The geostrophic velocities (Fig. 8) are surface intensified, which suggests that the surface drift should be larger than the DAC (which it is). The surface drift is influenced by ageostrophic flows such as the effect of the winds, whereas the DAC represents the average flow throughout the deeper water column (upper 1000 m).

L131: "south" – replace with "north"? Perhaps providing the horizontal scale of the bowl structure would help understanding.

Sorry for the confusing wording. The 'south' in this line refers to the location of the bowl structure, rather than the occurrence of warmer waters north of the Southern Boundary. Depending on the defined location of the Southern Boundary (southernmost limit of UCDW) the bowl structure would still be south of the Southern Boundary. This paragraph (now L156-164) has been edited to improve readability. Horizontal scale description (latitude) of the bowl structure has been added as well.

L133: What is "the coincident changes"?

The 'coincident changes' here refer to the characteristics, such as water mass properties and bowl structure south of the Southern Boundary that match for the

transects (A, B, D and E) which do not necessarily match for transect C. We have adapted the sentence as follows:

L274-279: In contrast, transect C demonstrates weaker horizontal density gradients in comparison to the other transects, which is implied by a less steeply sloping 27.93 $kg\,m^{-3}$ isopycnal. The 27.73 $kg\,m^{-3}$ isopycnal in transect C also does not bowl downwards 140 and does not show the changes in water mass properties, associated with the bowl-structure, as demonstrated in the other transects.

L143: "40 km" – the baroclinic deformation radius is 10-15km, then we can expect eddy's diameter of 20-30 km?
Yes, apologies for that. The eddies are about 20-30 km in diameter. We have edited the paper accordingly (L175).

L145: I could see westward velocities characteristic to the eddy's southern edge by the surface drift and the altimetric velocities, while they are unlikely visible in the DAC.
We apologize for that. The westward velocities at the eddy's southern edge are visible in the DAC too. We have adjusted the arrow size in Fig. 4 to increase visibility.

L148: "advected" – it might also be possible that the eddy was merged with a larger structure (probably, jet's meander) to its west or east.
Yes, absolutely. We have added your suggestion to L180-181.

L150: Then, how sea-level depressions (white contours) larger than the cold eddy can be interpreted?
We are not sure what the reviewer is referring to here? We are assuming that the reviewer is referring to Fig. 4. The other white contours here refer to other cold-core eddies interacting with the Southern Boundary. In Fig. 4 the white contours depict the transition zone from eddy core towards the outside of the eddy. We have added that explanation to the figure caption to clarify that.

L161: Absolute salinity needs unit g/kg.
Agreed, units have been added to absolute salinity throughout the manuscript.

L169: Why the DAC is more appropriate as the reference than the surface drift?
The DAC is a more appropriate reference for the geostrophic shear since this 0-1000 m average velocity will better represent the depth range over which the flow is predominantly geostrophic. The surface drift will be affected by the wind influence and other ageostrophic flows, so is less appropriate as a reference velocity.

L170: 80 cm/s – this far exceeds the altimetric speed and the surface drift.
Yes, this is quite a common issue. With regards to the surface drift please see L320-324. Satellite altimetry- derived currents are necessarily temporally and spatially smoothed by the process of creating the gridded product from relatively widely-spaced altimetric tracks infrequently repeated. This may lead to eddies and front being in the correct location, but averaged/smoothed in e.g. current speed so that values from satellite altimetry tend to be smaller than observed current speeds. We have added the following lines to the text to emphasize that in more detail.

L174: "the gradient of ADT (Fig. 8a,c)" – unit is m/m in Fig 8
Yes, thanks for spotting that. Unit has been corrected.

L177: It also seems like the major front (SACCF-N) and the minor front (SACCF-S) regulate the barrier strength. Can you please provide any effects by jet's meandering?
Only three of the observed transects (A,B and C) extend far enough south to encounter the minor jet and we only have data along one line of longitude. We believe questions around the effect on the frontal jet properties of the meandering path of the jets (e.g. the effect of latitudinal separation of the jets and the effect of jet curvature) would require a much more widespread study to address them, covering a longer time period and greater geographical extent. In fact, a modeling study would probably be more appropriate than an observational one. Given the limitations of our data set, we do not feel it would be appropriate for us to comment on the effect of the jet meandering.

L184: "strengthens" – does this refers to inverse cascade dynamically?
Yes, it does. However, we decided not to introduce this terminology to the paper as energy cascades are not the focus of this work.

L204: How the temperature fluctuation is calculated? (This would be why I could not fully understand the calculation)
We have added the following lines (L240-244) to the manuscript to clarify the calculation of $\Theta_{rms}$. Please see response to major point 2 for further details.

L203: Strictly speaking, the cross-section (defined by glider positions), along-stream (defined by the streamline), and zonal components are all different. Please elaborate on it throughout the manuscript or demonstrate these differences do not change the result.
First the glider locations are projected onto a meridional line and then we calculate the geostrophic shear. Thus, we only calculate the zonal velocity component. Furthermore, we find in the key transects (A and C) the flow at the Southern Boundary and over the associated frontal jet is zonal. This information has been added to the manuscript (L93-102).

L219: "The PV is further considered along potential density surfaces with…" – Simply, "PV is calculated over"? Or, is this meant to be "potential density surfaces are considered to be
isoneutral"?
According to the reviewer's suggestion the sentence has been changed to:
L259: The PV is mapped on potential density surfaces with…

L263: There is section 4.1 but following sections 4.2 etc. are absent.
We thank the reviewer for spotting this error. Section 4 is now divided into section 4.1 and section 4.2.

L275: "The Southern Boundary's location (determined from the frontal jet)" – I recommend to replace with the SACCF.
Please see response to major comment 1.

L287: "In summary" – meridional eddy heat flux may be given by $-k\nabla\Theta$, where k is isopycnal diffusivity associated with the mixing length. Then, how changes in $\nabla\Theta$ affect the
meridional heat transport? Is it safely negligible even on account of the offshore warming?
We have deleted this sentence from the paper, as we agree with the reviewer that we were overstating our results. Although k has decreased over the last decade, $\nabla\Theta$ has increased. We do not have numerical values for either the change in k or the change in $\nabla\Theta$, so we cannot be certain whether $k\nabla\Theta$ has increased or decreased.

Responses to Reviewers

Stirring across the Antarctic Circumpolar Current's Southern Boundary at the Greenwich Meridian, Weddell Sea

We thank the reviewer for their helpful comments and suggestions that have strengthened our paper. In our responses below, the reviewers' comments are in black, our responses are in blue and the revised text is in purple.

Reviewer 2

5 repeated glider surveys across the Southern Boundary (SBDY) of Antarctic Circumpolar Current are used to investigate SBDY's cross-frontal behaviours under eddy and non-eddy regimes. Eddy presence enhances cross-frontal density gradient supressing the cross-frontal mixing whereas eddy absence, comparing to eddy presence, is accompanied by a weaker cross-frontal density gradient. These results are interpreted under the context of a multidecadal evolution of SBDY speed/location derived from satellite data. Authors concluded that the enhanced eddy activities and accelerated SBDY are occurring at the same time in opposition in affecting the meridional exchanges of tracers cross SBDY at Greenwich Meridian. I found this work is interesting and potentially important for the community in understanding the Weddell Gyre heat content evolution under the context of climate changes.

I have one concern about this manuscript. This work highlights that the different cross-fontal properties are associated with eddy presences exemplified by comparing transect A and transect C. These contrasting results between eddy and non-eddy regimes need to be strengthened by a quantified uncertainty that could be raised from different glider sampling intensity along the transect because it seems to me that the transect C does not take profiles as frequently as transect A by looking at the station distribution from two transects. See also the relevant comments below. I am happy to see this manuscript published once my concerns herein are addressed properly.

We thank the reviewer for their positive comments and for the suggestion to strengthen the uncertainty. The reviewer is correct that different transects had different profile densities since this depends on the speed of the glider. We have addressed this through subsampling the more densely-sampled Transect A to match the sampling density of Transect C. We have run this subsampled transect through the same processing path. The results are discussed below.

General comments:

1. Most results present in this manuscript based on the comparison between transect A and transect C, where the authors argue that eddy presence/absence is the reason for the observe difference. The glider station (marked as triangle on top of cross-section plot, most evident in Figure 8) distribution between A and C is different. Can author quantify the potential uncertainty caused by different glider station distributions on the present cross-frontal difference?

Yes, the reviewer is correct that the data are sampled in a higher horizontal resolution in Transect A (290 vertical profiles over 320 km, 145 dives), whereas Transect C has a lower horizontal resolution (92 vertical profiles over 200 km, 46 dives). We have addressed this through subsampling of the more densely-sampled Transect A to the sampling density of Transect C. The subsampling was accomplished by firstly shortening Transect A to the meridional extent of Transect C and then by only including every other profile. We have run this subsampled transect through the same processing path. The results of the subsampled transect are shown in Fig. 1 below.

[Figure]

Fig. 1 (a) The mean temperature field $\Theta_m$, (b) the measure of the temperature fluctuations $\Theta_{rms}$, (c) the gradient of the mean temperature $\nabla_\rho \Theta_m$ along potential density surfaces, (d) the mixing length scales $L_{mix}$ for the subsampled transect A. All panels are spatially smoothed by a 30 km x 0.08 $kg\,m^{-3}$ moving median filter. All subfigures a-d are shown in density space with a vertical gridding of 0.02 $kg\,m^{-3}$.

Both the subsampled Transect A (Fig. 1 in this response) and the densely-sampled Transect A (Fig. 9a-d in the manuscript) show the same characteristics as described in the manuscript (reduced $L_{mix}$, strong gradients along isopycnals). To further visualise the differences by subsampling Transect A, we show the anomalies (Transect A - Transect A subsampled, Fig. 2 this response). Please note that specifically for the area of interest (Southern Boundary, 55.5-56°S) $L_{mix}$ and thus our key results remain unchanged (Fig. 2d).

[Figure]

Fig. 2 Anomalies (Transect A - Transect A subsampled) of (a) The mean temperature field $\Theta_m$, (b) the measure of the temperature fluctuations $\Theta_{rms}$, (c) the gradient of the mean temperature $\nabla_\rho \Theta_m$ along potential density surfaces, (d) the mixing length scales $L_{mix}$ for the subsampled transect A. All panels are spatially smoothed by a 30 km x 0.08 $kg\,m^{-3}$ moving median filter. All subfigures a-d are shown in density space with a vertical gridding of 0.02 $kg\,m^{-3}$.

Please note further that the data, for the mixing length scale diagnostics and calculation of geostrophic velocities, are vertically and horizontally gridded to achieve a uniform grid for all transects. We have added this information to the manuscript (L94-103) and changed the diamonds at the top of Fig. 8 to show that the grid for Transect A and C is uniform.

[Figure]

Figure 8. Real-time altimetric ADT and gradients of ADT ($\nabla y$ ADT) for (a) transect A and (c) transect C. (b,d) Geostrophic velocities perpendicular to the respective glider transects A and C and referenced to the DAC with a horizontal smoothing (moving mean filter) of approx. 15 km (Rossby radius within the region of interest). Positive geostrophic velocities are defined as eastwards (red). Black contours are as in Fig. 2. The black diamonds at the top of each panel show the uniform horizontal gridding with 5 km spacing of transect A and C. The dashed black lines indicate the location of the Southern Boundary based on the southernmost strong ADT gradient.

2. This may or may not be resolved by typesetting, but I found that quite a few figures are far from where they were discussed. For example, section 3 in page 8 discussed Figure 4 to Figure 8, while Figure 8 is displayed at Page 14. I suggest authors to condense down figure volume, such as, putting multiple subpanels into one integrated figure, leaving the results for transect B, D, E in Supp Mats as they were barely mentioned, T-S plots with highlighted regimes taking up one subpanel spaces can be replaced by combining mainly discussed regimes in one T-S plot and mask other data points with grey colour, etc.

We have now brought the figures as closely together as possible with our limited options. Further editing of the figure positions is a matter of typesetting of the final article. We will raise your concerns if the figures are still poorly

placed when we receive the proof of the article. With respect to merging subpanels of several figures, we must respectfully disagree. We consider a visual separation of figures concerning Transects A and C a helpful tool for the reader to enhance clarity. With respect to Figs. 1 and 2 we think that transects B, D, E should still be included to introduce the entire data set and to justify why we focus on Transects A and C later.

Specific comments:

L6: 'quite rapid'→'high-frequency' or 'transient'?
Quite rapid has been changed to transient (now L7).

L35: delete 'globally', the word 'globally' is misplaced as the SBDY is not a global feature, is it? 'Climatologically' is sufficient here.
This has been changed as suggested by the reviewer.

L42: '…further represent the southernmost boundary to mixing'. I found this sentence a bit ambiguous… I believe that the mixing process in general is happening everywhere, and I don't think authors have set the context of using the term mixing to refer the cross-frontal mixing happened at the SBDY.
This sentence has been edited according to the reviewer's concern. The sentence now reads as (L41-44): The frontal jets of the ACC are often seen as barriers to meridional horizontal mixing (e.g. Naveira Garabato et al. (2011)). The frontal jet associated with the Southern Boundary, as the southernmost of the ACC frontal jets, marks the boundary between the northern limit of sea ice formation and the ACC.

L60:'The majority of studies almost entirely…', need refs here or is author referring to aforementioned studies? If so, please indicate.
Yes, all aforementioned studies are referred to here. This has been edited in the manuscript as suggested by the reviewer.

L160: 'converge'. The T-S plots do not show this 'convergence' particularly clear. Adding arrows to indicate this in the T-S plots.
Arrows have been added to Figures 6 and 7 according to the reviewer's suggestion, and the convergence of temperatures and salinities is identified in the T-S plots (Figs. 6 and 7), representing UCDW, and is described in L117-131.

L164-167: I do not fully understand this. The similarity of the properties between eddy and south of SBDY suggest eddy originated from south of SBDY, okay, then what is the meaning of mentioning the slight temperature/salinity difference below/above the thermocline? Plus, why do authors mention the salinity difference in reference to thermocline?
We have adjusted the sentence to clarify (now L200-204):
The clockwise eddy identified in transect A (Fig. 6 a,d,e) presents properties similar to the cold regime but with slightly higher temperatures (about 0.4 to 0.6◦C higher) below the thermocline and slightly reduced salinities above the depth of the thermocline. Note that the eddy is surface intensified and therefore changes in the surface properties are expected, although the eddy is more clearly identified in the sub-thermocline temperatures and salinities. The similar water mass properties of the

eddy and the cold regime suggest that the eddy originated south of the Southern Boundary.

L179-182: Mention the criteria and the table somewhere earlier in the section. This section has covered many figures that use such color-coding criteria. Best to mention it in the first place to avoid confusion for readers.
We agree with the reviewer and have moved the table with the criteria earlier in the section to improve clarity. The criteria are now first discussed and explained, and the Table cited, in lines 191-195.

L185: the eddy passage could be one of the reasons for the difference in horizontal density gradient between transect A and C. Figure 8 shows a smooth ADT for C and rough ADT for A which makes me realize that the profiling intensity of A and C is also different. Transect A has more profiles in general than Transect C across the front. Does this fact play any role? Authors should quantify the uncertainty on horizontal density gradient caused by different sampling intensity by subsampling a high-res model results/reanalysis or any other sensible measures.
We have tested the subsampling of transect A as suggested. It did not impact the results for transect A and key characteristics of transect A do not change.  Please see response to major point 1 for further detail.

L204: It is not clearly stated how the temperature fluctuation, θ', is computed.
We have added the following lines (L240-244) to the manuscript to clarify the calculation of $\Theta_{rms}$: Finally, at each grid point we find the root mean square difference $\Theta_{rms}$ between the value of $\Theta_m$ at that grid point, and the values of $\Theta_\rho$ within a 5-element window in the horizontal (i.e., on the same density surface) centered on that grid point.  In other words,

$$\Theta_{rms,i} = \sqrt{\frac{\sum_{j=i-2}^{j=i+2} (\Theta_{\rho,j} - \Theta_{m,i})^2}{5}}$$

where $i$ is an index from south to north along a potential density surface.

L285: The discussion on the long-term behavior of the SBDY and its core speed is sufficiently supported by literatures. However, the sea ice extent seems to be a bit out of place here. I suggest authors to either specify the reason for examining sea ice extent and discuss it extensively in the context of past literatures or simply not to show the sea ice extent at all since it does not correlate well with the available data here and authors just briefly mentioned it… Sea ice advancing and retreats on yearly basis is also controlled by large-scale wind variability, thermal forcing and also internal sea ice dynamic, so it perhaps requires some extra effort to decipher sea ice extent in the context of enhanced frontal jet.
According to the suggestion  of the reviewer we have removed the sea ice extent from Fig. 11 as there is currently not enough literature to support our findings and the lack of correlation between frontal jet speed and sea ice extent does not provide enough evidence to further discuss the sea ice extent within this manuscript. We have further removed L44-45 and L285 from the manuscript.

L294: If authors are referring to the positive SLA blobs into the 2010s, then perhaps the phrase 'anti-cyclonic eddies' is more appropriate than warm core eddies? Studies

have shown that not all anti-cyclonic eddies have a coherent warm core structure throughout the vertical extent.
The expression warm core eddy has been replaced with anticyclonic eddy (L336).

L309: '…. consistent with Williams et al. (2007) who demonstrated …'
Edited according to the reviewers suggestion (L351).

L301: '… in all transects …', authors mainly discussed transects A, relevant results for B, D, E should be included at least in Supp Mats to make this claim.
Relevant results for B, D and E have been added to the Appendix for completeness (page 27-29).